# UniCode: A Framework for Generating High-Quality Competitive Coding Problems

## Abstract

The reliance of competitive coding benchmarks on static, human-authored problems creates significant challenges, including data contamination and limited scalability. To address these issues, we introduce **UniCode**, a novel framework that automatically generates high-quality algorithmic problems alongside robust, contamination-resistant test cases. Inspired by biological evolution that creates better and diverse offspring, our framework leverages Large Language Models (LLMs) to systematically diversify problems through three strategies: single-problem extension, same-type fusion, and cross-type fusion. A key innovation is our stress-driven test case synthesis pipeline, which generates reliable test suites without requiring a canonical ground-truth solution. This pipeline combines brute-force grounding for small-scale inputs with a consensus-based validation mechanism for large-scale inputs to ensure high correctness and coverage. We demonstrate effectiveness of our framework by curating a benchmark of 492 problems and evaluating 19 state-of-the-art LLMs. The results reveal that Uni-Code is highly challenging and discriminative, with the top-performing model, o4-mini, achieving a pass rate of only 70.3%. Our framework provides a scalable and reliable solution for generating dynamic evaluation datasets in coding domain.

## 1 Introduction

The rapid advancement of large language models (LLMs) in code generation (Li et al., 2022; Allal et al., 2023; Jaech et al., 2024; Guo et al., 2024; Hui et al., 2024; Comanici et al., 2025; Guo et al., 2025; Zhao et al., 2025) presents an evaluation paradox. While models are approaching saturation on existing static benchmarks (Zhu et al., 2025; Chen et al., 2021; Austin et al., 2021; Hendrycks et al., 2021), it is increasingly uncertain whether this performance reflects genuine reasoning ability or simply vast memorization of training data (Oren et al., 2023; Roberts et al., 2023; Golchin & Surdeanu, 2023; Riddell et al., 2024; Tang et al., 2024; Li et al., 2025). Even advanced benchmarks that periodically add new problems, like LiveCodeBench (Jain et al., 2024) and LiveCodeBench-Pro (Zheng et al., 2025b), are constrained by their reliance on costly, slow human authoring, which limits their scalability. This core issue is rendering static evaluation obsolete and demands a fundamental shift in methodology. We argue that the next frontier lies in generative evaluation (Zhu et al., 2023; Lin et al., 2025; Parmar et al., 2024; Shi et al., 2025; Zheng et al., 2025a): a paradigm where the benchmark itself is an intelligent system, dynamically producing a near-endless stream of novel challenges to probe the true generalization capabilities of models.

To meet this challenge, we introduce UniCode, a generative evaluation framework that automatically synthesizes novel, competition-level programming problems and robust test suites at scale. At the heart of our framework is a problem generation engine inspired by biological evolution (Kutschera & Niklas, 2004; Pagel, 1999; Koonin, 2011). The diversity of life is driven by two key mechanisms: gene mutation and genetic recombination (Baake & Gabriel, 2000; Charlesworth et al., 2009). Similarly, UniCode employs three complementary strategies to create new tasks from a seed set (as shown in Figure 5): single-problem extension, which modifies a problem to increase its complexity (e.g., evolving Two Sum to Three Sum); same-type fusion, which combines two problems with the same algorithmic tag to create a new variant with similar logic but a different narrative; and cross-type fusion, which merges problems from different categories to form complex hybrid challenges (e.g., Palindrome Sum Pair). This methodology can produce well-formed and structurally novel problems, with a solvability rate of 98.2% confirmed by blind human review.

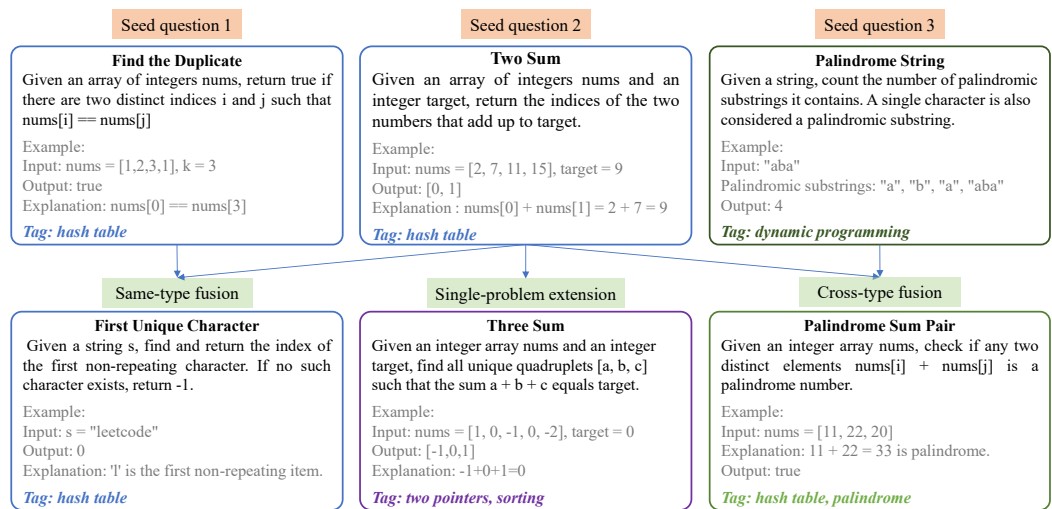

Figure 1: An illustration of our three problem generation strategies using seed problems. **Same-type fusion** (left): combines two problems sharing a common tag (e.g., hash table) to produce a new problem with the same core logic but greatly different descriptions. **Single-problem extension** (middle): modifies a seed problem to create a more complex version (e.g., *Two Sum → Three sum*). **Cross-type fusion** (right): merges problems from different categories to create a hybrid challenge (e.g., *Palindrome Sum Pair*). For concrete examples of generated problems, please refer to Appendix section G.

A key innovation of UniCode is its stress-driven test case synthesis pipeline (Figure 2), which ensures the quality and reliability of the evaluation without needing a canonical ground-truth solution. This process first establishes trusted outputs for small-scale inputs using a brute-force solver, which are then used to "stress-test" and filter a pool of LLM-generated optimized solutions. For large-scale inputs, where brute force is infeasible, correctness is determined by a majority-vote consensus among the filtered, high-quality solvers, with a powerful LLM adjudicating disputes to enhance reliability. Our experiments demonstrate that this pipeline achieves a test case correctness rate of 94.5% while also providing robust coverage that effectively identifies flaws in imperfect solutions.

We demonstrate our framework's effectiveness by curating a new benchmark of 492 problems and evaluating 19 state-of-the-art LLMs. Our results confirm that UniCode is highly challenging and discriminative: the top-performing model, o4-mini, achieves a pass@1 rate of only 66.1%. A controlled experiment confirms that our generation strategy effectively produces novel and challenging problems derived from data-contamination seeds, as models exhibit an average performance drop of over 30% on UniCode problems compared to the original seed set (see Figure 3). The evaluation also reveals that leading open-source models are highly competitive, signaling a promising direction for reproducible and accessible AI research. In summary, UniCode provides a scalable, reliable, and contamination-resistant solution for the dynamic evaluation of competitive coding.

## 2 METHODOLOGY

This section details the UniCode framework, which systematically generates novel competitive programming problems and their corresponding robust test suites. Our approach is divided into two core pipelines: problem generation (Section 2.1) and test case generation (Section 2.2).

### 2.1 PROBLEM GENERATION

The key idea behind generating new problems is to modify existing ones. We first observe that naive superficial changes to problems, such as changing the background story or adding distracting information, do not produce challenging or novel tasks, as confirmed by our experiments (Figure 3). Therefore, we design generative process that are more powerful to create diverse problems for our framework. Let $\mathcal{D}$ be a dataset of programming problems, where each problem is tagged with one or more algorithm types from a set $\mathcal{T}$. To generate a new problem, $p_{\text{new}}$, we sample one or two seed problems from $\mathcal{D}$ and apply one of three strategies:

**Single-problem extension**  This strategy creates a new problem by modifying a single seed problem $p_a$. We provide an LLM with $p_a$ and a `modification instruction` $\phi$. These instructions can include actions like tightening constraints, altering the input format, or adding a side condition. The LLM then generates a new problem $p_{\text{new}}$, that retains the core structure of the original but introduces a fresh challenge. For example, the LLM can evolve the classic *Two Sum* problem into *Three sum*, a more complex variant requiring a different algorithmic approach (Figure 5, bottom middle).

**Same-type fusion**  This strategy generates a novel problem by fusing two seed problems $p_a$ and $p_b$, that share a common algorithmic tag $t$ (e.g., *hash-table*). The LLM is prompted to first identify the underlying solution pattern shared by both problems and then instantiate this pattern into a new problem $p_{\text{new}}$, with a distinct narrative but the same core solution logic. For example, by analyzing two different problems that both use hash tables for efficient lookups, the model can generate a new frequency-counting task that leverages the same core data structure (Figure 5, bottom left). Providing concrete seed problems as in-context examples, rather than just a tag $t$, is crucial. It helps constrain the LLM to generate new problems of comparable quality and difficulty, which leads to a greater diversity of more appropriate challenges.

**Cross-type fusion**  This strategy creates more difficult, hybrid problems that require integrating multiple skills. We start with two seed problems $p_a$ and $p_b$, from `different` algorithmic categories. The LLM identifies a "bridging concept"—a compatible element that can link the two problems. This concept is then used to integrate a feature from one problem into the other, generating a composite problem $p_{\text{new}}$. For example, merging a *Two Sum* problem (hash table) with a *Palindrome checking* problem (strings, dynamic programming) could result in a new *Palindrome Sum Pair* task, which requires a solution that combines both skills (Figure 5 bottom right).

## 2.2  Test Case Generation

Generating reliable test cases for novel problems without a known ground-truth solution is a significant challenge. We address this with a two-part pipeline that first generates diverse inputs and then establishes trusted outputs through a hybrid brute-force and consensus-based approach.

### 2.2.1  Input Generation

Based on previous work in test case generation (Jain et al., 2024; Liu et al., 2025; Wang et al., 2025), we construct a comprehensive test suite for each problem by prompting LLMs to generate inputs from three complementary sources:

1. **Random Generation** ($G_{\text{rand}}$)**:** A Python function that samples broadly from the valid input space to ensure wide coverage, including both small-cale and large-scale instances.

2. **Adversarial Generation** ($G_{\text{adv}}$)**:** A Python function that targets boundary conditions and worst-case scenarios using predefined strategies (e.g., maximum values, alternating values) to probe algorithmic robustness and efficiency.

3. **LLM-based Synthesis** ($G_{\text{llm}}$)**:** This method synthesizes full input text, offering greater flexibility without external libraries. It produces small-scale yet challenging inputs designed to expose subtle, hard-to-find failure modes, and is categorized as adversarial in our experiments.

All candidate inputs, regardless of their source, are validated by the verifier $V$. For each input $\tau \in \{\text{rand}, \text{adv}, \text{llm}\}$, let $G_\tau$ be the candidate pool. The final set of verified inputs $I_\tau$ is defined as:

$$I_\tau = \{\, x \in G_\tau \mid V(x) = \text{true} \,\}.$$

To balance coverage and efficiency (Liu et al., 2023a), we assemble a final test suite $S$ of 50 cases with a fixed composition: 20 random, 20 adversarial, and 10 LLM-synthesized inputs, with the parameters determined through an empirical sweep (see Appendix Section D). We discuss in Table 1 to show this setting has good correctness and coverage.

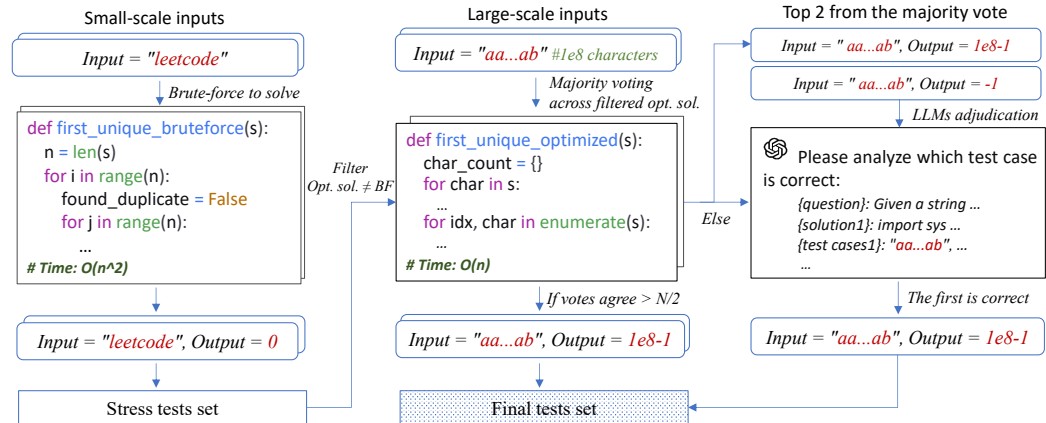

Figure 2: Stress-driven pipeline for ground-truth generation. **(1) Stress Testing:** A brute-force solver establishes trusted outputs for small-scale inputs. These are used as a "stress test" to filter a pool of LLM-generated optimized solutions. **(2) Consensus Validation:** The filtered, efficient solvers are run on large-scale inputs, and outputs are determined by a strict majority vote. **(3) LLM Adjudication:** A powerful LLM adjudicates between conflicting outputs for inputs where no majority is reached.

### 2.2.2 OUTPUT GENERATION

Establishing ground-truth outputs for novel problems is challenging. We devise a multi-stage pipeline (Figure 2) that mirrors rigorous human validation practices, separating inputs into *small-scale* and *large-scale* sets to balance reliability with computational feasibility.

**Stage 1: Brute-Force Validation and Solver Filtration.** For small-scale inputs ($I_s$), where correctness can be verified tractably, we first generate a **brute-force solver** ($B$) via an LLM prompt. This solver, while inefficient, is optimized for correctness. We execute $B$ on all inputs in $I_s$ to create a trusted set of ground-truth pairs, forming our initial stress test suite.[1]

Next, we prompt various LLMs to produce a diverse set of $M$ optimized candidate solutions, $\{C_1, \ldots, C_M\}$. Each candidate $C_j$ is then validated against our stress test suite. A solution is admitted to our trusted solver pool $\mathcal{P}$ only if it correctly reproduces the brute-force output for every small-scale input:

$$\mathcal{P} = \{C_j \mid C_j(i) = B(i) \text{ for all } i \in I_s\}.$$

This filtration step ensures that only solutions proven correct on a baseline set of inputs are used for subsequent stages, yielding a pool of $N = |\mathcal{P}|$ efficient and pre-vetted solvers.

**Stage 2: Consensus Validation on Large-Scale Inputs.** For large-scale inputs ($I_\ell$), which include all test cases generated by random, adversarial, and LLM-based synthesis methods, executing a brute-force solver is computationally infeasible (see Appendix section C). Instead, we leverage the trusted pool $\mathcal{P}$ of efficient solvers. For each input $i \in I_\ell$, we execute all $N$ solvers in $\mathcal{P}$ and collect their outputs. The final ground-truth output is determined by a **strict majority vote**: an output $o$ is accepted if it is produced by more than $\lfloor N/2 \rfloor$ solvers.

**Stage 3: LLM Adjudication for Disagreements.** If no single output achieves a strict majority, the input is flagged for adjudication. In this case, we identify the top two most frequent outputs, $o_1$ and $o_2$, and present them to a powerful LLM (e.g., o4-mini). The LLM is tasked with analyzing the problem statement, the input, and the conflicting candidate solutions to determine which output is correct. If the LLM provides a decisive judgment, its chosen output is accepted as the ground truth. If the conflict cannot be resolved, the input is discarded from the final test set to maintain its integrity. This final step allows us to resolve complex edge cases that might stump a purely algorithmic consensus. We validate each component in this pipeline can improve the accuracy of test case in Table 1.

---

[1]To mitigate the risk of a single faulty brute-force solver, we generate multiple candidates and use their consensus output, further enhancing reliability.

# 3 BENCHMARK CURATION

This section describes how we construct the benchmark produced by our UniCode framework and how we validate the resulting problems and test suites. We first detail the data sources and curation pipeline used to assemble the benchmark, then report a human study assessing problem quality and clarity.

## 3.1 BENCHMARK STRUCTURE

**Seed problems.** We start from the TACO dataset, which aggregates over $25k$ programming problems with human-written solutions from platforms e.g., Codeforces, LeetCode, and CodeChef.

**Filtering for competitive difficulty.** We filter the dataset by: (i) removing non-competitive problems; (ii) eliminating duplicates and problems with unclear input/output specifications.

**Target coverage and tag taxonomy.** From the filtered pool, we target the 15 most prevalent algorithmic tags to ensure breadth while keeping consistent difficulty. Our tag set includes hierarchical refinements (e.g., *graph → shortest-paths*, *flow*; *dp → knapsack*, *interval*). The full taxonomy and mapping rules appear in Appendix E. The tags are assigned by an LLM, which we prompt to identify the 1-3 most relevant skills for each problem (see Appendix I).

**Problem generation and selection.** We adopt the *o4-mini-medium* to generate a set of candidate problems based on our initial problem pool, using the UniCode methods described in §2. Each generated problem is paired with a test suite averaging 50 cases. To ensure the problems are challenging, we filter out any that can be solved perfectly by *all* our baseline models. Concretely, we run a panel of baselines as in Table 2 if every model achieves a $100\%$ pass rate on the test suite, the problem is removed. Our final benchmark contains 492 problems spanning the 15 target tags.

**Test suites and constraints.** Each problem includes five components: description $D$, tag set $\mathcal{T}$, time limit $TL$, memory limit $ML$, and test cases $M$. Test cases are assembled as in §2.2 with a fixed composition of $|I_{\text{rand}}| = 20$, $|I_{\text{adv}}| = 20$, and $|I_{\text{llm}}| = 10$ (total $|M| = 50$). The time limit $TL$ and memory limit $ML$ are determined by running validated, optimized solutions $\mathcal{O}_{\text{valid}}$:

$$\text{TL} = \left\lceil k \cdot \min_{o \in \mathcal{O}_{\text{valid}}} T(o) \right\rceil, \qquad \text{ML} = \left\lceil k \cdot \text{Mem}(o^\star) \right\rceil,$$

where $o^\star = \arg\min_{o \in \mathcal{O}_{\text{valid}}} T(o)$. We set safety factor $k = 3$, and execute $T(\cdot)$ and $\text{Mem}(\cdot)$ within a secure sandbox environment (Bytedance-seed et al., 2025). To facilitate analysis, we also store per-case provenance (generator type, random seed, and verifier decisions).

## 3.2 QUALITY OF GENERATED PROBLEMS

**Human evaluation protocol.** We assess clarity and solvability through a blinded rating study (see Appendix F). Five experienced annotators (competitive programmers/algorithm engineers, $\geq 5$ years experience) independently rate a uniformly random sample of 113 problems from the 492-problem benchmark. We provide each rater with the problem statement and example I/Os. The primary metric is binary *solvability* (solvable / unsolvable): whether the statement is unambiguous and admits a well-defined solution within the stated constraints.

**Results.** Across the 113 evaluated problems, the solvability rate is 98.2%. For detailed human label metrics, see Appendix F. The average inter-annotator agreement rate with human assessments is 92.3%, indicating that the problem statements were clear and well-posed for expert readers.

**Release artifacts.** We will release problem statements, test suites (with provenance), and metadata (tags, generators, and prompts) to support reproducibility and downstream analysis.

Table 1: Quality of generated test suites. We compare our full pipeline against the rStar-Coder baseline and ablations of our own method. **Stage 1 (Brute-Force)** uses brute-force solvers on small random inputs. **MajVote (Unfiltered Solvers)** applies a majority vote to all generated solutions without pre-filtering. **MajVote (Filtered Solvers)** uses only solutions that passed the Stage 1 stress test. Our **Full Pipeline** integrates all stages and input types. "Validated suites" is the percentage of problems for which a suite passed all validation checks.

| | Method | Input types | Correctness | Coverage | Problems with validated suites (%) |
|---|---|---|---|---|---|
| rStar-Coder (Liu et al., 2025) | *MajVote (Unfiltered)* | *Rand* | 86.9% | 80.2% | 94.3% |
| Ours | *Stage 1 (Brute-Force)* | $Rand_s$ | 91.9% | 81.5% | 98.2% |
| | *MajVote (Unfiltered)* | $Rand_l$ & *Adv* | 86.7% | 85.2% | 93.9% |
| | *MajVote (Filtered)* | $Rand_l$ & *Adv* | 93.8% | 84.3% | 92.8% |
| | ***Full Pipeline*** | *Rand* **&** *Adv* | **94.5%** | **86.0%** | **94.8%** |

## 3.3 EVALUATING TEST CASE QUALITY

We evaluate test suites using two metrics: *correctness* (accepting valid solutions) and *coverage* (rejecting invalid ones). A robust test suite must excel at both. Let $S_{\text{correct}}$ and $S_{\text{incorrect}}$ be sets of correct and incorrect submissions. A test suite $M$ passes a submission $s$ if $s$ succeeds on all test cases $m_i \in M$, denoted $\text{pass}(s, M) = 1$. We define:

$$\text{Corr@N} = \frac{|\{\, s \in S_{\text{correct}} \mid \text{pass}(s, M) = 1 \,\}|}{|S_{\text{correct}}|}, \quad \text{Cov@N} = \frac{|\{\, s \in S_{\text{incorrect}} \mid \text{pass}(s, M) = 0 \,\}|}{|S_{\text{incorrect}}|}.$$

**Setup.** We test on 80 problems from the *Test-Eval dataset* (Yang et al., 2025), using suites of size $N = 50$. Note that our approach to majority voting differs from rStar-Coder in its granularity. While rStar-Coder aggregates at the solution level and discards a problem unless a majority of solutions exhibit identical input–output behavior; our aggregation is performed per test case, so disagreement on individual cases does not invalidate the entire problem.

**Results.** Our full pipeline significantly outperforms the baseline, achieving higher scores in both correctness (94.5% vs. 86.9%) and coverage (86.0% vs. 80.2%). We attribute this improvement to several key factors. Our Stage 1 filtering boosts correctness from 86.7% to 93.8% by using brute-force solvers on small inputs to eliminate flawed solutions. At the same time, we improve coverage by using adversarial inputs to uncover edge-case failures that are often missed by random inputs. Although our system does not achieve 100% accuracy, the mathematical proof in Appendix B demonstrates that the benchmark's reported accuracy is still trustworthy. Finally, the validated-suite rate slightly exceeds the baseline because our method of per-test-case adjudication helps us retain reliable, yet difficult problems that would otherwise be discarded.

## 4 EVALUATING LLMS ON ALGORITHMIC PROBLEMS

This section presents the experimental setup and results of our evaluation of large language models (LLMs) on the UniCode benchmark. We analyze model performance across various difficulty levels and compare the capabilities of different model series, providing insights into their strengths and weaknesses on code generation tasks.

## 4.1 EXPERIMENTAL SETUP

**Models** We evaluated 19 diverse large language models, including open-source and closed-source models with varied architectures and sizes. Our selection includes models optimized for reasoning, instruction-following, and code generation. Key models are **GPT series** (e.g., `gpt-4.1-mini`, `gpt-4o`), **Gemini series** (`gemini-2.5-pro`, `gemini-2.5-flash`), **Claude series** (`claude-3.7-sonnet`, `claude-sonnet-4`). Other competitive models from the **DeepSeek** and **Qwen series**, as well as `llama-4-maverick` and `gemma-3-27b`, are also included for a comprehensive comparison. We use a unified prompt with a model temperature of 0.2 across all models.

| Model | Easy | Medium | Hard | Overall | RandPass | AdvPass | $\Delta(A-R)$ |
|---|---|---|---|---|---|---|---|
| o4-mini (high)* | 94.9% | 78.2% | 21.6% | 70.3% | 80.1% | 72.7% | -7.4% |
| gpt-5 (medium)* | 89.5% | 77.6% | 18.8% | 67.7% | 75.9% | 70.0% | -5.9% |
| o4-mini (medium)* | 89.2% | 73.6% | 20.3% | 66.1% | 80.4% | 67.5% | -12.9% |
| google/gemini-2.5-pro* | 94.0% | 53.1% | 8.5% | 61.6% | 74.7% | 64.3% | -10.4% |
| deepseek-v3.1 | 89.2% | 59.8% | 11.5% | 60.5% | 70.3% | 64.4% | -5.9% |
| deepseek-r1 | 80.3% | 36.4% | 5.1% | 55.6% | 69.6% | 59.6% | -10.0% |
| o3-mini (medium)* | 86.2% | 50.0% | 6.0% | 55.1% | 66.8% | 57.5% | -9.3% |
| qwen3-235b-a22b | 80.2% | 39.7% | 5.1% | 53.5% | 68.2% | 56.0% | -12.2% |
| gemini-2.5-flash* | 81.4% | 22.6% | 4.8% | 47.7% | 61.8% | 50.4% | -11.4% |
| grok-3-mini* | 77.8% | 21.7% | 3.3% | 46.4% | 61.8% | 48.3% | -13.5% |
| claude-3.7-sonnet* | 76.2% | 24.1% | 2.4% | 45.5% | 63.9% | 44.5% | -19.4% |
| deepseek-chat-v3.1 | 82.7% | 29.3% | 3.9% | 49.8% | 61.0% | 52.0% | -9.0% |
| gpt-4.1-mini* | 73.7% | 20.9% | 3.8% | 42.4% | 58.1% | 45.4% | -12.7% |
| gpt-4.1* | 62.1% | 21.8% | 1.4% | 36.5% | 51.8% | 39.4% | -12.4% |
| qwen3-coder | 66.5% | 9.3% | 0.0% | 35.4% | 53.9% | 36.9% | -17.0% |
| claude-sonnet-4* | 60.7% | 14.0% | 2.0% | 32.4% | 48.3% | 34.1% | -14.2% |
| llama-4-maverick | 51.3% | 8.6% | 0.0% | 26.2% | 41.3% | 27.5% | -13.8% |
| gpt-4o* | 31.3% | 2.2% | 0.0% | 15.4% | 23.9% | 15.2% | -8.7% |
| qwen-2.5-32b-coder | 27.2% | 2.2% | 0.0% | 13.4% | 26.8% | 11.5% | -15.3% |
| gemma-3-27b-it | 26.1% | 2.2% | 0.0% | 13.1% | 21.2% | 14.0% | -7.2% |
| llama-3.3-8b-instruct | 11.2% | 1.1% | 0.0% | 5.5% | 12.8% | 5.8% | -7.0% |

Table 2: Pass@1 performance on the UniCode benchmark. The table is split by reasoning (upper) and non-reasoning (lower) models. Closed models are marked with *. Models are ranked by overall average Pass@1 in descending order. RandPass and AdvPass denote pass rates on random and adversarial test cases respectively. $\Delta(A-R)$ denotes the gap between AdvPass and RandPass.

## 4.2 PASS@1 PERFORMANCE OVERVIEW

We report pass@1 results on the UniCode benchmark, which clearly ranks models by performance and difficulty levels while exposing their brittle failure modes. Our key findings are:

**UniCode is challenging and highly discriminative.** Even the top model, *o4-mini-high*, attains an overall pass@1 of 70.3%, while the weakest model (*llama-3.3-8b-instruct*) reaches only 5.5%, a significant gap of 64.8% points. Performance collapses sharply on hard problems: several models, such as *gpt-4o* and *llama-4-maverick*, record 0.0% on the hard split, underscoring the benchmark's capacity to distinguish multi-step algorithmic competence. The results further reveal that reasoning-oriented models consistently outperform others. The top six reasoning models achieve an average pass@1 of 57.3%, which is more than double the 30.0% average of the non-reasoning models. This substantial gap highlights the value of architectures and fine-tuning specifically for structured, multi-step reasoning in competitive coding tasks. The ranking results from the UniCode benchmark show over 90% alignment with other popular code benchmarks, as detailed in Appendix A.3

**Models consistently struggle with challenging test cases.** We contrast performance on randomly sampled inputs (RandPass) with adversarially generated inputs (AdvPass). Across all models, the average drop from RandPass to AdvPass is 11.2% points, indicating widespread brittleness to edge cases. The most significant declines occur in *claude-3.7-sonnet* with a 19.4% point decrease and *qwen3-coder* with a 17.0% point decrease. Interestingly, while *o4-mini-medium* achieves a Rand-Pass rate similar to *o4-mini-high*, it exhibits a substantial 5.2% point gap in AdvPass at 67.5% versus 72.7%. This disparity suggests that higher-capacity reasoning models possess stronger abilities to handle complex corner cases, thereby improving adversarial robustness.

**Competitive open-source models are on the rise.** While closed-source models generally lead in overall performance, a select few open-source models are proving to be serious contenders. Models like *DeepSeek-v3.1* and *Qwen3-235b-a22b* demonstrate robust capabilities, showcasing the immense potential of the open-source community. Notably, *DeepSeek-chat-v3.1*, a non-reasoning model, achieves a overall pass@1 of 49.8%, outperforming several reasoning-oriented models like *gemini-2.5-flash*. The strong performance of these open-source models provides immense value to the research community. They offer transparency and scientific understanding, which allows re-

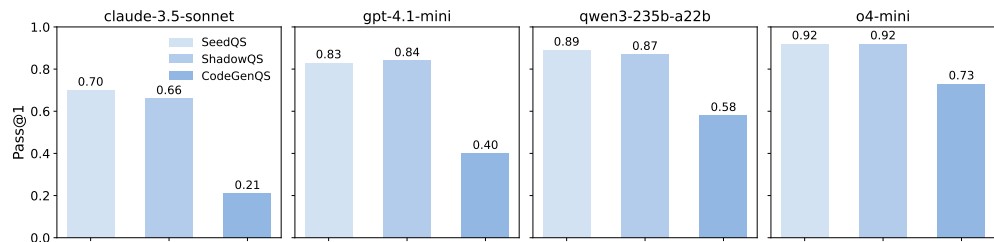

Figure 3: Performance comparison across question sets. *SeedQS* denotes the original seed questions, *ShadowQS* represents the semantically equivalent shadow questions generated from *SeedQS*, and *CodeGenQS* refers to the novel question set created by our *UniCode* method.

searchers to conduct deeper investigations into model architecture, training data, and limitations, fostering innovation and accelerating progress in competitive programming.

We further explore the pass@k performance in Appendix A.2. The performance for *o4-mini-medium* rises from $66.1\%(k = 1)$ to $83.5\%(k = 10)$ but plateaus after $k = 7$, indicating diminishing returns. The final pass rate underscores the benchmark's challenge even with extensive sampling.

## 4.3 Evaluating Generalization in LLMs

In this section, we conduct data-contamination experiments to evaluate generation strategies capable of producing genuinely novel problems and mitigating model memorization, then examine the generalization ability of models that can flexibly apply learned knowledge to solve novel problems. We construct three problem sets derived from 50 problems in LiveCodeBench v1 [2], an early version with potential data contamination risk. **Seed questions**: the original 20 problems. **Shadow questions**: reformulated versions with altered wording, added stories, or distracting details, while preserving the same logical core. These generation strategies include transformations such as reframing a card-game queue simulation as an operating-system scheduling scenario (changing the narrative while keeping the queue-management logic intact), embedding a simple algorithmic task within a longer story, or adding irrelevant tables and conditions that leave the underlying computation unchanged. **UniCode questions**: new problems generated by UniCode (section 2.1), requiring transfer and flexible use of seed knowledge. Each dataset contains 48 problems. We evaluated pass@1 performance across four LLMs, yielding key insights into their generalization abilities:

**LLMs Are Robust to Superficial Changes** Most LLMs perform nearly identically on seed and shadow questions. This suggests a robust capacity to generalize across variations in problem descriptions. Models effectively distill the core logical structure, remaining insensitive to superficial textual or narrative modifications. The nearly unchanged performance on shadow questions confirms that simple text-based transformations do not defend against data contamination.

**Algorithmic Generalization Remains a Challenge** All LLMs exhibited a significant performance drop on UniCode questions. The decline was particularly pronounced in *claude-3.5-sonnet* (0.70 → 0.21) and *gpt-4.1-mini* (0.83 → 0.40). This indicates that while models handle narrative variations well, they struggle when faced with problems requiring novel or composite algorithmic reasoning. *Qwen3-235b-a22b* and *o4-mini-medium* demonstrated relatively stronger generalization, suggesting a more flexible problem-solving architecture. The dramatic performance drop on UniCode questions demonstrates that UniCode successfully generates novel algorithmic problems that cannot be solved through memorization or pattern matching, even when seed problems may have been contaminated.

## 4.4 Performance Across Algorithmic Paradigms

To better understand model capabilities, we sample representative models and categorize the problems by their primary algorithmic paradigm. Performance is evaluated using the pass@1 metric. As shown in Figure 4, the results reveal distinct strengths and weaknesses across various problem types.

**Strengths in template-driven tasks, weaknesses in complex reasoning.** The models demonstrate high proficiency in deterministic, template-driven tasks such as *fundamentals* and *data struc-*

---

[2]Initial Release: May 2023 - March 2024.

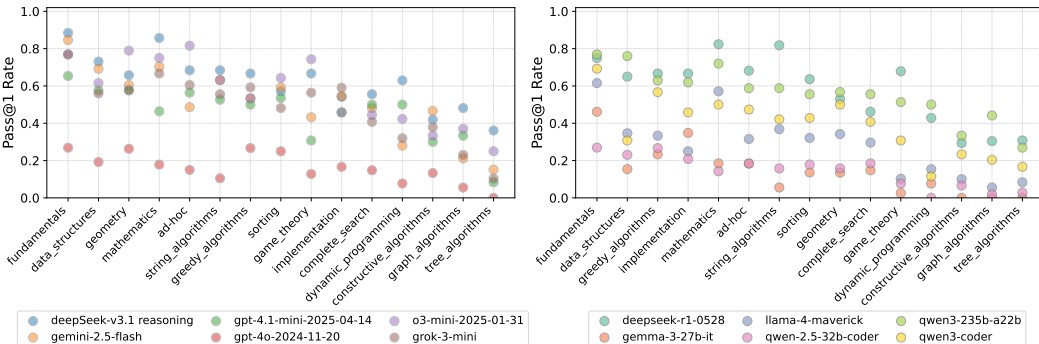

Figure 4: Comparative performance across problem types for closed-source (left) and open-source (right) models. The x-axis represents algorithmic tags. The y-axis refers to the pass@1 rate. Best viewed zoomed in.

*tures*. These problems, which often involve standard data structure manipulations, are likely well-represented in training corpora from textbooks and online repositories. Their solutions typically follow predictable patterns that models can easily recognize and reproduce. In contrast, performance drops significantly on problems requiring novel reasoning and multi-step planning, such as *graph algorithms* and *dynamic programming* problems. These domains often require customized logical deduction rather than simple pattern matching. This performance gap suggests that current models may rely more on memorization of common solutions than on robust reasoning.

## 4.5 ANALYSIS OF GENERATOR BIAS

A potential concern in generative evaluation is the risk of "generator bias," where a model performs better on problems it generated itself. To address this, we performed validation using an alternative, open-source generator `deepseek-r1` to generate a new set of 104 problems across 5 distinct tags. We then benchmark 6 models of varying capability levels on this independently generated set and compared the results to their performance on the standard `o4-mini`-generated UniCode problems and human-curated no data contamination LiveCodeBench[3].

| Model | Unicode (deepseek-generated) | Unicode (o4mini-generated) | LiveCodeBench (human-curated, no contamination) |
|-------|------------------------------|----------------------------|------------------------------------------------|
| *gpt-5* | 72.5% | 67.7% | — |
| *o4-mini* | 70.2% | 66.1% | 74.2% |
| *deepseek-r1* | 61.6% | 55.6% | 73.1% |
| *o3-mini* | 51.0% | 55.1% | 63.0% |
| *gpt-4.1-mini* | 41.3% | 42.4% | 53.2% |
| *gemma-3-27b-it* | 14.6% | 13.1% | — |

Table 3: Pass@1 rates across different problem generators. While absolute scores show fluctuation across the three datasets, indicating differences in generator difficulty or stylistic preferences. The relative model hierarchy remains highly consistent (Pearson $r = 0.984$ between `deepseek-r1` and `o4-mini` generated sets).

As indicated in Table 3, the results do not support the presence of significant self-preference bias. The relative ranking of models remains highly consistent across all three datasets, with a Pearson correlation of $r = 0.984$ ($p = 4 \times 10^{-4}$) between model performances on the `deepseek-r1`-generated and `o4-mini`-generated problem sets.

Although `deepseek-r1` performs markedly better on its own problems than on `o4-mini`'s (61.6% vs. 55.6%), a similar performance gap persists on human-curated LiveCodeBench. This suggests the difference stems from stylistic preferences inherent to each generator, rather than intentional bias. Therefore, to mitigate any residual stylistic bias and ensure robust evaluation, we will release UniCode-Multi benchmark, a composite benchmark aggregating 100 problems each from `o4-mini`, `gpt-5`, `gemini-2.5-pro`, `Deepseek-r1`, and `qwen3-235b-a22b` models.

---

[3]Initial Release: 8/1/2024 to 5/1/2025

## 5 RELATED WORK

### 5.1 COMPETITIVE CODE GENERATION

Evaluating the code generation capabilities of large language models is a rapidly evolving field, driven by the proliferation of powerful models (Jaech et al., 2024; Li et al., 2023; Guo et al., 2024; Hui et al., 2024; Zhang et al., 2023; Guo et al., 2025; Li et al., 2022; Shao et al., 2024). Foundational benchmarks like HumanEval (Chen et al., 2021), MBPP (Austin et al., 2021), and APPS (Hendrycks et al., 2021) established evaluation standards based on functional correctness on standalone programming puzzles. However, their static nature and limited difficulty spectrum make them susceptible to data contamination (Oren et al., 2023; Golchin & Surdeanu, 2023; Riddell et al., 2024; Roberts et al., 2023), where models may have seen solutions during pretraining. To address this, subsequent benchmarks (Li et al.; Gu et al.; Zhu et al., 2025; Chambon et al.) draw from competitive programming to assess more complex reasoning, and some mitigate contamination with recent contest problems (Zheng et al., 2025b; Jain et al., 2024), yet updates remain slow. Our work introduces a scalable framework that generates a infinite stream of novel, competitive coding problems.

### 5.2 AUTOMATED TEST CASE GENERATION

Generating comprehensive test cases for accurate assessment of algorithmic correctness is difficult to scale because it relies on manual curation (Chen et al., 2021; Hendrycks et al., 2021; Austin et al., 2021; Li et al.; Quan et al., 2025). Recent efforts leverage large language models for test generation (Chen et al., 2022; Schäfer et al., 2023; Liu et al., 2023a), with frameworks like Code-Contests++ (Wang et al., 2025) and LiveCodeBench (Jain et al., 2024) employing this for competitive programming problems. However, these methods often crucially depend on existing solution code (Schäfer et al., 2023; Tufano et al., 2022; Chen et al., 2022; Liu et al., 2023b), making them unsuitable for novel problems. AutoCodeBench (Chou et al., 2025) attempts to solve this by generating problems from existing code; however, the resulting tasks can lack algorithmic complexity. Our framework overcomes these limitations with a stress-driven synthesis pipeline that generates high-quality test cases for novel problems without needing a reference solution.

### 5.3 GENERATIVE EVALUATION

Generative evaluation aims to overcome benchmark stagnation and data contamination (Oren et al., 2023; Roberts et al., 2023) by dynamically creating novel tasks. This paradigm is explored in open-ended game environments, where MCU (Zheng et al., 2025a) combined atomic tasks into complex new ones, and KUMO (Lin et al., 2025) extended this to practical domains like medical diagnosis. Other works apply generative methods to logical reasoning (Parmar et al., 2024), dynamic evaluation on synthetic data structures (Zhu et al., 2023) or agentic tasks (Shi et al., 2025) to achieve more comprehensive analysis. Our work adapts the principle of generative evaluation to assess complex algorithmic code generation. We create diverse coding problems by fusion existing ones (Pei et al., 2025; Huang et al., 2025; Wu et al., 2021) and generating variations to measure true generalization. While methods like Evol-Instruct (Xu et al., 2024) and WizardCoder (Luo et al., 2023) evolve problem descriptions to enhance instruction-following, UniCode instead targets the evolution of the underlying algorithmic structure, generating problems that require genuinely novel reasoning.

## 6 CONCLUSION

In this paper, we introduced UniCode, a new framework designed to generate high-quality competitive programming problems. Our key contributions are a biological evolution-inspired generation strategy and a stress-driven test case synthesis pipeline. By combining these approaches, UniCode generates unique and challenging problems alongside robust, high-coverage test suites. We validated UniCode by evaluating 19 state-of-the-art LLMs, confirming its effectiveness as a challenging and highly discriminative benchmark. This work represents a significant step forward in our ability to evaluate generative models in the demanding field of competitive programming. A limitation of our study is its reliance on a single powerful closed-source LLM, *o4-mini*, which may introduce bias by generating problems that align with its own strengths.Future work will investigate employing diverse LLMs as problem generators and developing more cost-effective methods.

## ETHICS STATEMENT

The authors have carefully considered the ethical implications of this work. This work on automated problem generation is guided by the imperative to contribute to society and human well-being by advancing the fair and robust evaluation of AI systems. We have considered potential societal impacts and strived to uphold the highest standards of scientific integrity, transparency, and fairness throughout the research process. Our methodology is designed to avoid harm by promoting the development of more reliable and generalizable AI models, and we have taken care to respect intellectual property and the creative work inherent in algorithmic problem design. This work aligns with the goal of ensuring that progress in AI is measured ethically and contributes positively to the field.

## REPRODUCIBILITY STATEMENT

To support reproducibility, we will publicly release the UniCode benchmark, including all 492 generated problems, test suites with provenance metadata, and detailed prompts used for problem and test case generation. The experimental setup—including model versions, evaluation metrics, and hyperparameters is clearly documented in the paper. Code for the stress driven test synthesis pipeline and problem generation strategies will be provided as supplementary material. Human evaluation protocols and rating guidelines are described in Appendix. All constraints, input/output formats, and tagging taxonomies are fully specified to enable exact replication. We encourage the community to build upon this dynamically generated benchmark for future evaluations.

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

# A SUPPLEMENTARY EXPERIMENTAL RESULTS

## A.1 PERFORMANCE AND COST

This section analyzes the trade-offs between model performance, inference cost, and output verbosity. Figure 5 presents a Unicode leaderboard, plotting the Pass@1 score against the average cost per problem for various models. This visualization highlights a key finding: a higher Pass@1 score often correlates with increased average cost. For example, models like o4-mini (high) and gpt-5 (medium), which are top performers, also incur a relatively higher cost. This chart provides a clear visual representation of the performance-cost efficiency frontier, helping users select models that balance their budget and performance requirements.

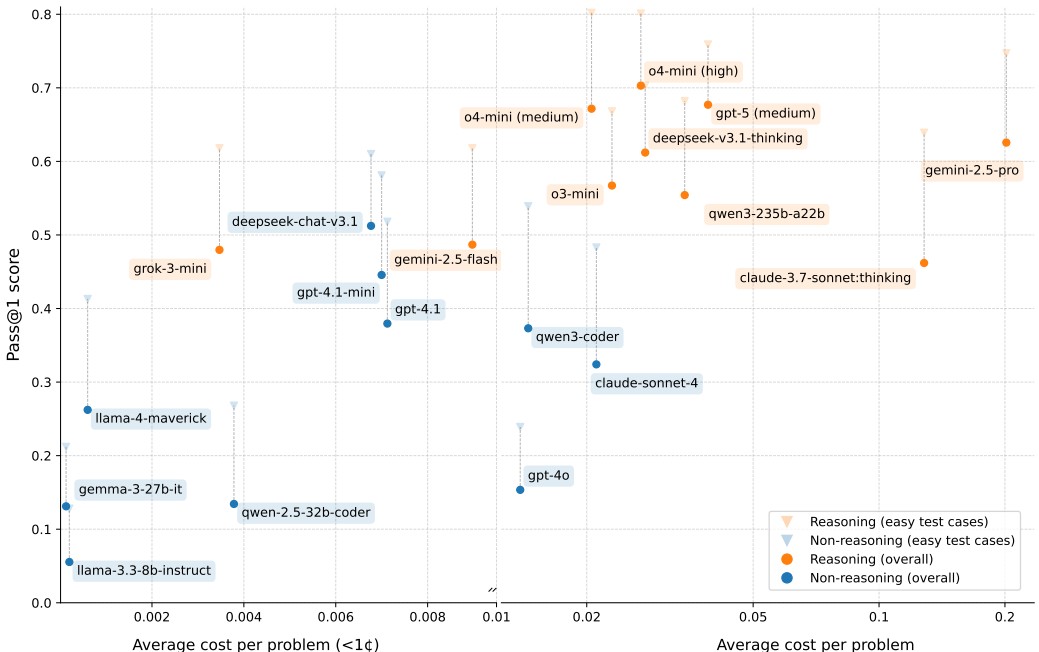

Figure 5: Unicode leaderboard. Pass@1 score vs. average cost per problem across various models.

Table 4 provides a detailed breakdown of these metrics. We report the Overall Pass@1 score, the empirical average inference cost per query (in US dollars), and the average number of output tokens. Models are ranked in descending order of their Overall Pass@1 score to emphasize performance. The table is structured to allow for easy comparison between performance, cost, and output length. We have included two distinct groups: reasoning models (top block), which are specifically designed for complex problem-solving, and general-purpose models (bottom block). The specific versions of the models are as follows: *gpt-5-2025-08-07, o4-mini-2025-04-16 high, o4-mini-2025-04-16 medium, gemini-2.5-pro, deepseek-v3.1-thinking, deepseek-r1-0528, o3-mini-2025-01-31, qwen3-235b-a22b, gemini-2.5-flash, grok-3-mini, claude-3.7-sonnet:thinking, deepseek-chat-v3.1, gpt-4.1-mini-2025-04-14, gpt-4.1-2025-04-14, qwen3-coder, claude-sonnet-4-20250514, llama-4-maverick:free, gpt-4o-2024-11-20, qwen-2.5-32b-coder*, and *llama-3.3-8b-instruct*.

## A.2 PERFORMANCE ACROSS PASS@K SETTINGS

Since we generate the dataset using `o4-mini-medium`, we study the *pass@k* performance directly on it. *Pass@k* metric defines a problem as solved if any of the top-$k$ generated candidates passes all test cases. The results are shown in Table 5.

The pass rate rises from 66.1% at $k = 1$ to 83.5% at $k = 10$, an improvement of over 20 percentage points. This indicates that repeated attempts significantly enhance performance, suggesting that many problems require multiple diverse generations to solve. The performance plateaus after $k = 8$, implying diminishing returns beyond this point. Even with multiple attempts, the model does not

| Model | Overall | Avg Cost ($) | Avg Output Tokens |
|---|---|---|---|
| *o4-mini (high)*[*] | 70.3% | 0.0269 | 6016.8 |
| *gpt-5 (medium)*[*] | 67.7% | 0.0390 | 3940.9 |
| *o4-mini (medium)*[*] | 66.1% | 0.0205 | 4565.2 |
| *google/gemini-2.5-pro*[*] | 61.6% | 0.2015 | 20099.6 |
| *deepseek-v3.1 (thinking)* | 60.5% | 0.0276 | 15990.0 |
| *deepseek-r1* | 55.6% | 0.0250 | 11337.5 |
| *o3-mini (medium)* | 55.1% | 0.0230 | 4782.0 |
| *qwen3-235b-a22b* | 53.5% | 0.0343 | 4046.0 |
| *gemini-2.5-flash*[*] | 47.7% | 0.0090 | 3249.6 |
| *grok-3-mini*[*] | 46.4% | 0.0035 | 6136.75 |
| *claude-3.7-sonnet*[*] | 45.5% | 0.1282 | 16856.8 |
| *deepseek-chat-v3.1* | 49.8% | 0.0068 | 3062.5 |
| *gpt-4.1-mini*[*] | 42.4% | 0.0070 | 3193.2 |
| *gpt-4.1*[*] | 36.5% | 0.0071 | 877.6 |
| *qwen3-coder* | 35.4% | 0.0145 | 2577.2 |
| *claude-sonnet-4*[*] | 32.4% | 0.0211 | 1319.8 |
| *llama-4-maverick (free)* | 26.2% | 0.0006 | 1211.4 |
| *gpt-4o*[*] | 15.4% | 0.0139 | 904.2 |
| *qwen-2.5-32b-coder* | 13.4% | 0.0038 | 898.6 |
| *llama-3.3-8b-instruct* | 5.5% | 0.0002 | 399.4 |

Table 4: Model ranking by Overall pass@1 (left) with per-model average inference cost and average output length (right). Reasoning models are listed first; closed-source models are marked with an asterisk. Overall percentages are reproduced from the reference table and used to rank models in descending order.

Table 5: `o4-mini (medium)` performance for *pass@k* settings

| k | 1 | 2 | 3 | 4 | 5 | 6 | 7 | 8 | 9 | 10 |
|---|---|---|---|---|---|---|---|---|---|---|
| **Pass Rate (%)** | 66.1 | 73.6 | 77.4 | 79.8 | 81.4 | 82.5 | 82.8 | 83.5 | 83.5 | 83.5 |

achieve a near-perfect score, illustrating the benchmark has high upper limit and diagnostic value. These results underscore the importance of sampling numerous candidates for difficult tasks and reflect the complexity and variability inherent in the problems.

To analyze the difficulty and recoverability of different problem types, we classified algorithm tags into three groups based on the improvement in pass rates from pass@1 to pass@3: Significant Improvement ($\geq$20%): This group includes tags like tree algorithms, graph algorithms, and mathematical problems. These problems often have multiple correct solutions or implementation approaches, so generating multiple code candidates significantly increases the pass rate. Moderate Improvement (10%–20%):: This group contains tags such as string algorithms and data structures. These problems typically have a more defined algorithmic template, and multiple samples can partially mitigate implementation errors. Minor Improvement ($\leq$10%): This group includes tags like dynamic programming and greedy algorithms. The limited effect of increasing samples is due to either a high initial pass rate or a reliance on structural reasoning rather than random variations. The disparity in improvement across tags indicates that multi-candidate evaluation is highly effective for problems with diverse solution pathways, but yields diminishing returns for tasks requiring deep structural insight or systematic implementation correctness.

## A.3 ALIGN WITH OTHER CODE BENCHMARKS

To validate UniCode, we conducted a Pearson correlation analysis with LiveCodeBench and LiveCodeBenchPro, using a consistent set of models. As depicted in Figure 7, we observed a strong positive correlation ($r > 0.9$) with LiveCodeBench. Conversely, a strong negative correlation ($r < -0.9$) is found with LiveCodeBenchPro. This negative correlation arises because LiveCodeBenchPro uses ranking-based scores where smaller values indicate better performance, whereas our metric follows a convention in which larger values correspond to better performance. This demonstrates that UniCode provides a consistent and reliable measure of model performance, aligning with established benchmarks.

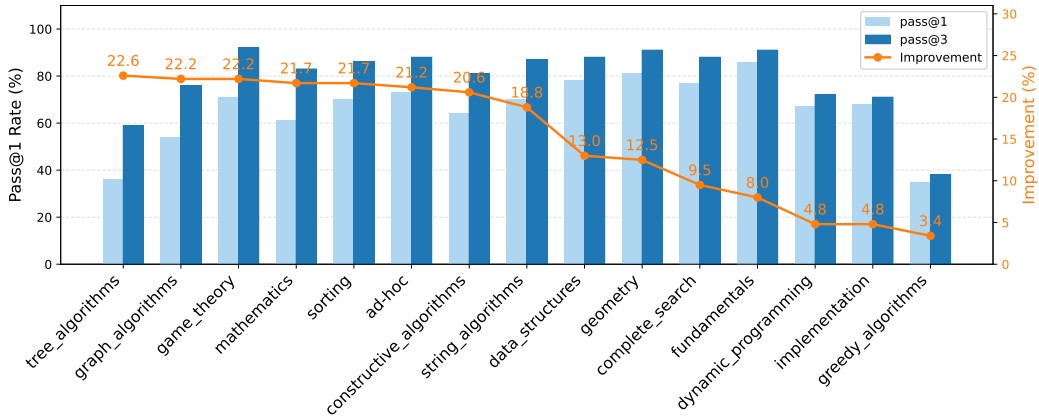

Figure 6: Per-tag improvement from *pass@1* to *pass@3*. Tags are grouped by improvement magnitude to illustrate which problem classes benefit most from candidate diversification.

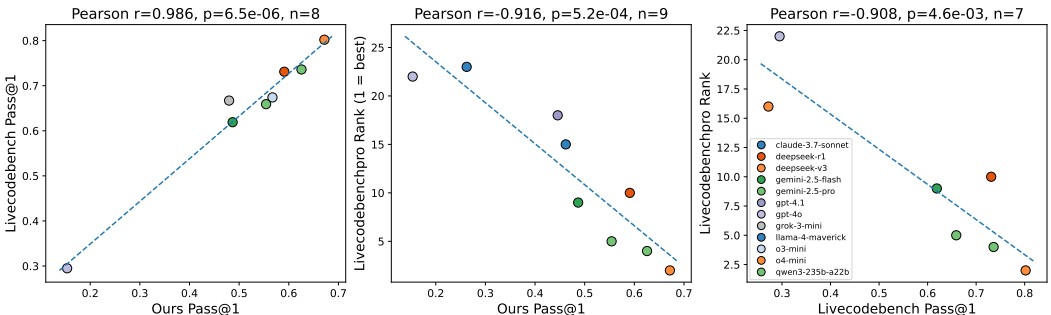

Figure 7: The alignment between UniCode and established benchmarks (LiveCodeBench and LiveCodeBench-Pro). The degree of alignment we achieve, with reference to the absolute value of the correlation coefficient $r$, surpasses the inter-correlation among the established benchmarks.

# B  TRUSTWORTHY EVALUATION WITH ERRONEOUS TASKS

Benchmarks for code generation occasionally contain *erroneous* items (e.g., unsolvable prompts, mislabeled I/O, flawed tests). This section develops a simple contamination model that quantifies how such items affect reported accuracy, provides bias- and variance-aware confidence bounds, and gives practical recipes to maintain trust in benchmark results.

## B.0.1  SETUP AND NOTATION

Let each task $i \in \{1, \ldots, n\}$ be either *reliable* ($R_i = 1$) or *unreliable* ($R_i = 0$). Write

$$\alpha \equiv \Pr(R_i = 0) \quad \text{and} \quad 1 - \alpha \equiv \Pr(R_i = 1),$$

so $\alpha$ is the *contamination rate* of the benchmark. Let $p \in [0, 1]$ denote the model's true accuracy on reliable tasks and let $q_e \in [0, 1]$ denote the effective success probability on unreliable tasks (e.g., a random or spurious pass rate). For *pass@k*, define $p$ and $q_e$ analogously as the success probability within $k$ attempts.

For each task, the observed outcome $Y_i \in \{0, 1\}$ indicates success. The reported accuracy is $\hat{\mu} \equiv \frac{1}{n} \sum_{i=1}^{n} Y_i$.

## B.0.2  SYSTEMATIC BIAS (IDENTIFICATION AND DE-BIASING)

By the law of total expectation,

$$\mathbb{E}[\hat{\mu}] = (1 - \alpha) p + \alpha q_e \quad \Longrightarrow \quad \text{Bias}(\hat{\mu}; p) = \left| \mathbb{E}[\hat{\mu}] - p \right| = \alpha \left| q_e - p \right| \leq \alpha. \tag{1}$$

Thus, when $\alpha$ is small, the systematic bias is small in absolute value. If $\alpha$ and $q_e$ are known (or fixed by design), an *unbiased* estimator of $p$ is obtained by de-biasing:

$$\tilde{p} = \frac{\hat{\mu} - \alpha q_e}{1 - \alpha} \qquad \text{(exact if } \alpha, q_e \text{ are known).} \tag{2}$$

When only bounds are available, $q_e \in [q_{\min}, q_{\max}]$ and $\alpha \in [0, \alpha_{\max}]$, one obtains a *conservative* identification region for $p$:

$$p \in \left[ \frac{\hat{\mu} - \alpha_{\max} q_{\max}}{1 - \alpha_{\max}}, \; \frac{\hat{\mu} - \alpha_{\min} q_{\min}}{1 - \alpha_{\min}} \right] \cap [0, 1]. \tag{3}$$

In practice, $q_{\max}$ can be set by a null-model baseline (e.g., trivial solver or random program generator), and $\alpha_{\max}$ by audit sampling.

### B.0.3 RANDOM ERROR (SAMPLING VARIABILITY)

There are two natural regimes for variance, depending on whether the reliable/unreliable split is fixed in advance (e.g., exactly $100(1-\alpha)\%$ reliable) or arises by i.i.d. sampling.

**Fixed split (common in controlled curation).** If exactly $(1-\alpha)n$ reliable and $\alpha n$ unreliable tasks are present,[4] then

$$\mathrm{Var}(\hat{\mu}) = \frac{(1-\alpha)\,p(1-p) + \alpha\,q_e(1-q_e)}{n}, \tag{4}$$

$$\mathrm{SE}(\hat{\mu}) = \sqrt{\frac{(1-\alpha)\,p(1-p) + \alpha\,q_e(1-q_e)}{n}}. \tag{5}$$

**Random mixture (i.i.d. contamination).** Marginally $Y_i \sim \mathrm{Bernoulli}(\mu)$ with $\mu = (1-\alpha)p + \alpha q_e$, hence

$$\mathrm{Var}(\hat{\mu}) = \frac{\mu(1-\mu)}{n} \qquad \text{with} \quad \mu = (1-\alpha)p + \alpha q_e. \tag{6}$$

By concavity of $x(1-x)$, equation 4 $\leq$ equation 6, so using equation 6 is conservative when the split is fixed.

**Confidence intervals.** Let $z_{0.975} \approx 1.96$. A simple large-sample 95% CI for $\mu$ is

$$\hat{\mu} \; \pm \; z_{0.975} \sqrt{\frac{\hat{\mu}(1-\hat{\mu})}{n}}, \tag{7}$$

or, more accurately at small $n$, use a Wilson or Agresti–Coull interval for $\mu$. When $\alpha, q_e$ are *known*, a CI for $p$ follows from de-biasing:

$$\left[ \frac{\underline{\mu} - \alpha q_e}{1 - \alpha}, \; \frac{\overline{\mu} - \alpha q_e}{1 - \alpha} \right], \tag{8}$$

where $[\underline{\mu}, \overline{\mu}]$ is a 95% CI for $\mu$. If only bounds are known ($\alpha \in [0, \alpha_{\max}]$, $q_e \in [q_{\min}, q_{\max}]$), combine equation 3 with $[\underline{\mu}, \overline{\mu}]$ to obtain a conservative CI for $p$:

$$\left[ \frac{\underline{\mu} - \alpha_{\max} q_{\max}}{1 - \alpha_{\max}}, \; \frac{\overline{\mu} - \alpha_{\min} q_{\min}}{1 - \alpha_{\min}} \right] \cap [0, 1]. \tag{9}$$

### B.0.4 TOTAL ERROR BOUND

Combining systematic and random components yields a high-probability bound on the absolute estimation error for $p$:

$$|\hat{\mu} - p| \; \leq \; \underbrace{\alpha\,|q_e - p|}_{\text{systematic bias } \leq \alpha} \; + \; \underbrace{z_{0.975} \sqrt{\frac{\mu(1-\mu)}{n}}}_{\text{random error}} \quad \text{(w.h.p.).} \tag{10}$$

When $n$ is large, the $O(n^{-1/2})$ term vanishes and the total error is controlled by the bias ceiling $\alpha$. If $\alpha$ (and $q_e$) are known, report the de-biased estimate equation 2 with CI equation 8; this both removes the bias and shrinks the CI.

---

[4] Assuming $n(1-\alpha)$ and $n\alpha$ are integers; otherwise interpret as the nearest integers.

### B.0.5 Stratified (Tag-wise) Contamination

If tasks are grouped into tags $t = 1, \ldots, T$ with weights $w_t$ (sum to 1), reliable rates $p_t$, contamination rates $\alpha_t$, and unreliable success $q_{e,t}$, then

$$\mu = \sum_{t=1}^{T} w_t\big((1 - \alpha_t)p_t + \alpha_t q_{e,t}\big), \tag{11}$$

$$\mathrm{Var}(\hat{\mu}) = \frac{1}{n}\sum_{t=1}^{T} w_t\big((1 - \alpha_t)p_t(1 - p_t) + \alpha_t q_{e,t}(1 - q_{e,t})\big), \tag{12}$$

under a fixed per-tag split. Reporting tag-wise de-biased estimates $\tilde{p}_t = (\hat{\mu}_t - \alpha_t q_{e,t})/(1 - \alpha_t)$ with their CIs, and then aggregating by the $w_t$, makes contamination assumptions explicit and auditable.

### B.0.6 Numerical Illustration

Take $\alpha = 0.06$, $p = 0.80$, $q_e = 0.50$. Then

$$\mu = (1 - \alpha)p + \alpha q_e = 0.94 \cdot 0.80 + 0.06 \cdot 0.50 = 0.782, \quad \mathrm{Bias} = |\mu - p| = 1.8\%.$$

Under a fixed split, the standard error is

$$\mathrm{SE}(\hat{\mu}) = \sqrt{\frac{0.94 \cdot 0.80 \cdot 0.20 + 0.06 \cdot 0.50 \cdot 0.50}{n}} = \sqrt{\frac{0.1654}{n}}.$$

The resulting 95% CI half-width is $1.96 \times \mathrm{SE}(\hat{\mu})$:

Table 6: Random error and conservative total error bound (bias + half-width) at various $n$ ($\alpha$=6%, $p$=0.80, $q_e$=0.50).

| $n$ | SE | 95% CI half-width | Total error bound |
|---|---|---|---|
| 500 | 1.82% | 3.57% | $1.8\% + 3.57\% \approx 5.4\%$ |
| 5,000 | 0.575% | 1.13% | $1.8\% + 1.13\% \approx 2.9\%$ |
| 10,000 | 0.407% | 0.80% | $1.8\% + 0.80\% \approx 2.6\%$ |

As $n$ grows, random error shrinks as $O(n^{-1/2})$; the residual error is then dominated by the (small) bias ceiling $\alpha$.

### B.0.7 Practical Safeguards

- **Audit and bound $\alpha$.** Spot-check a random subsample to obtain an empirical upper bound $\alpha_{\max}$ with binomial CIs; report $p$ using equation 9.
- **Calibrate $q_e$.** Measure $q_e$ (or $q_{\max}$) using null models (e.g., trivial programs, permuted I/O) to cap spurious pass rates.
- **De-bias when possible.** If $(\alpha, q_e)$ are fixed by design (e.g., known faulty items), publish the de-biased estimate equation 2 and its CI equation 8.
- **Stratify and reweight.** Estimate per-tag $(\alpha_t, q_{e,t})$ and aggregate, reducing sensitivity to heterogeneous contamination.
- **Robust reporting.** Alongside $\hat{\mu}$, report (i) de-biased $\tilde{p}$, (ii) contamination-aware CIs, and (iii) sensitivity bands under $(\alpha, q_e)$ ranges as in equation 9.

**Takeaway.** Even when a benchmark contains a small fraction of erroneous tasks, its reported accuracy remains trustworthy when (i) contamination is explicitly modeled, (ii) bias is de-biased or bounded, and (iii) sampling error is controlled by adequate $n$. In the common regime of small $\alpha$ and large $n$, the total measurement error is tightly bounded and the benchmark reliably reflects true coding performance.

# C ALGORITHM CAPACITY ANALYSIS

We analyze how algorithms of different time complexities behave under constrained time limits. Consider a programming competition problem requiring:

- **Task**: Sort an array of $n$ integers and count inversions
- **Input Range**: $1 \leq n \leq 2 \times 10^7$
- **Expected Solutions**:
    - Optimal: Merge sort ($O(n \log n)$) with inversion counting
    - Suboptimal: Bubble sort ($O(n^2)$) with brute-force counting

### C.0.1 CAPACITY ANALYSIS

We assume a typical modern computer can perform approximately $10^8$ operations per second. We set time limits as 5s for optimized and 50s for brute-force algorithms.

For the optimized algorithm ($T(n) = n \log_2 n$): we solve $n \log_2 n \leq 5 \times 10^8$ to show it can handle input $n = 2 \times 10^7$ in 5 seconds, as the number of operations, $2 \times 10^7 \times 24.2 \approx 4.84 \times 10^8$, stays within the $5 \times 10^8$ operations limit for 5 seconds at $10^8$ operations per second.

For the brute-force algorithm ($T(n) = n^2$): $n^2 \leq 50 \times 10^8$ yields $n \approx 7 \times 10^4$ maximum. In contrast, processing input $n = 2 \times 10^7$ would require $4 \times 10^{14}$ operations (around 46 days), demonstrating quadratic time growth.

Table 7: Algorithm Capacity Comparison

| Metric | Optimized ($O(n \log n)$) | Brute-force ($O(n^2)$) |
|---|---|---|
| Time Limit | 5s | 50s |
| Max $n$ | $2 \times 10^7$ | $7 \times 10^4$ |

The large difference (approximately 300x) in manageable input sizes ($2 \times 10^7$ vs $7 \times 10^4$) explains the stress-driven pipeline: the optimized algorithm verifies efficiency at competition-scale inputs, while the brute-force method allows small-case validation ($n \leq 10^4$ in $\leq 2$s). This setting ensures that the brute-force algorithm has enough time to pass test cases with smaller input sizes, which are usually used to verify basic correctness. This is very useful for debugging and initial testing.

# D  TEST-SUITE COMPOSITION AND PARAMETER SELECTION

In this section, we provide a detailed justification for the composition of the final test suite $S$, which consists of 50 test cases with a fixed distribution: 20 random ($I_{\text{rand}}$), 20 adversarial ($I_{\text{adv}}$), and 10 LLM-synthesized ($I_{\text{llm}}$) inputs. This configuration was determined through an extensive empirical evaluation aimed at balancing correctness and coverage.

## D.1  EXPERIMENTAL SETUP

We conducted a hyperparameter sweep over multiple test suite compositions, evaluating each configuration on a held-out set of 48 problems with 960 human-crafted solutions. Each configuration was assessed using two key metrics:

- **Correctness**: the proportion of valid solutions that pass all test cases.
- **Coverage**: the proportion of invalid solutions that are correctly rejected.

Table 8: Representative sweep results across different distributions.

| Distribution | Correctness (%) | Coverage (%) |
|---|---|---|
| (5, 5, 0) | 97.9 | 77.0 |
| (10, 10, 5) | 95.8 | 81.4 |
| **(20, 20, 10)** | **94.0** | **87.5** |
| (30, 30, 20) | 91.7 | 88.7 |
| (50, 50, 20) | 91.7 | 90.0 |

We observe that smaller test suites (e.g., (5,5,0)) achieve high correctness but suffer from low coverage, failing to detect many faulty solutions. Larger suites (e.g., (30,30,20)) improve coverage marginally but at the cost of increased sensitivity to corner test cases and higher computational cost. The configuration (20, 20, 10) strikes an optimal balance, maintaining high correctness 94.0% while achieving strong coverage 87.5%, without excessive computational overhead.

## D.2  ABLATION ON CATEGORY NECESSITY

To validate the necessity of each test case category, we performed an ablation study by removing one category at a time from the full suite and measuring the impact on overall pass rates. The results are summarized in Table 9. LLM-synthesized test cases are labeled as "Corner" in the table.

Table 9: Comprehensive Model Evaluation with Ablation Study

| Model | Pass Rate (%) | | | Ablation Study (%) | | | Overall Pass (%) |
|---|---|---|---|---|---|---|---|
| | Adv | Random | Corner | -Adv | -Random | -Corner | |
| Average | 52.99 | 55.92 | 55.53 | 49.94 | 44.53 | 51.21 | 38.99 |

Removing any category leads to a consistent increase in the overall pass rate, indicating that each type contributes uniquely to fault detection. Adversarial cases are critical for detecting edge-case failures. Random inputs ensure broad coverage of the input space. LLM-synthesized cases expose subtle logical errors that are often missed by the other two types. Based on empirical sweeps and ablation evidence, we selected a final suite of 50 test cases with a fixed composition of 20 Random, 20 Adversarial, and 10 LLM-synthesized inputs.

## E  CODE-TAG DISTRIBUTION

To systematically evaluate code generation capabilities of large language models (LLMs), we constructed a hierarchical taxonomy that organizes algorithmic knowledge into tags, subtags, and atomic skills. In total, the taxonomy consists of **9 top-level tags**, **31 subtags**, and **161 skills**, covering both fundamental algorithms and advanced techniques. Figure 8 provides a summary of this distribution.

This taxonomy brings several advantages for code generation evaluation. First, it ensures broad algorithmic coverage: the tags span essential paradigms such as graph algorithms, dynamic programming, data structures, and mathematical methods, allowing evaluations to probe diverse coding skills. Second, the inclusion of fine-grained subtags and skills provides diagnostic granularity. Instead of producing only aggregate scores, we can profile model performance across different algorithmic domains, exposing specific strengths (e.g., string hashing, greedy heuristics) and weaknesses (e.g., bitmask dynamic programming, numerical stability). Third, tagged organization supports balanced dataset construction, ensuring that evaluations are not biased toward a narrow set of skills. It also facilitates longitudinal comparisons: since the taxonomy is stable, we can track progress across model iterations and architectures. Finally, many of the listed skills, such as hashing and network flow, are directly relevant to industrial software engineering and competitive programming, thereby improving the real-world applicability of the evaluation.

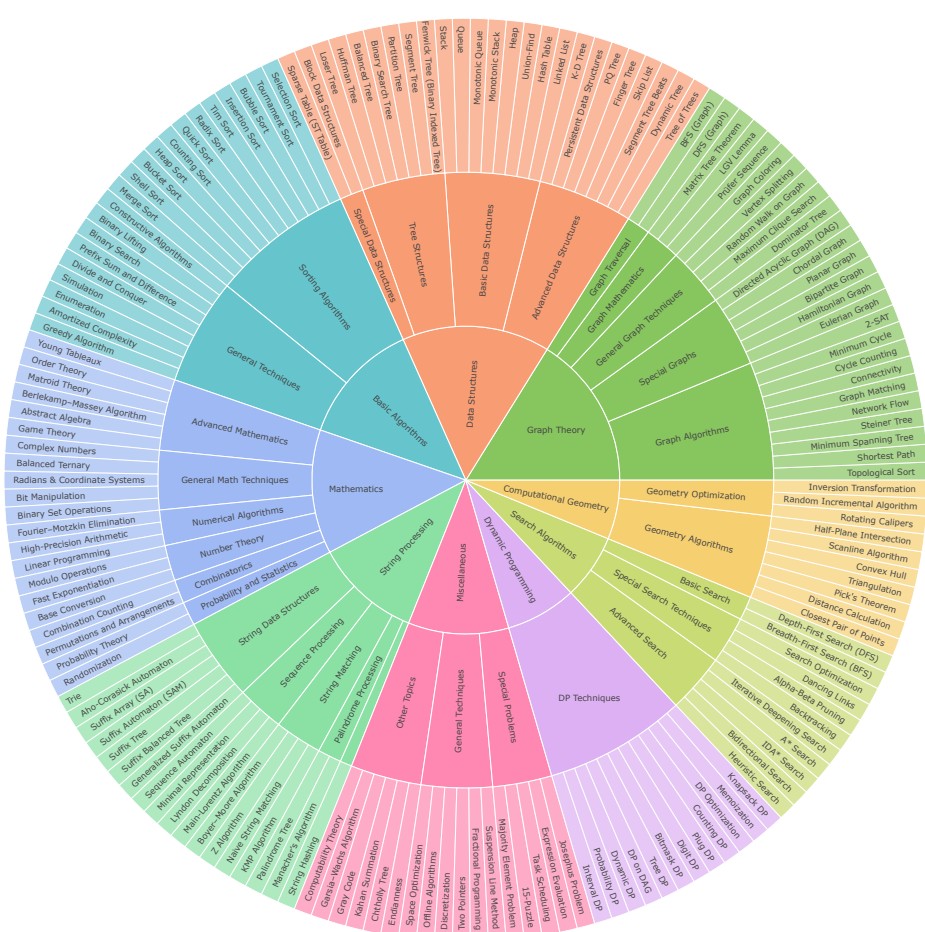

Figure 8: Distribution of tags, sub-tags, and skills in UniCode dataset.

# F  HUMAN EVALUATION OF GENERATED PROBLEMS

We conduct a comprehensive evaluation of the generated problems, bringing the total number of evaluated problems to 113. Each problem was reviewed by experienced competitive programmers. Every item, including the two original seed problems and the newly generated problems, is annotated according to the following criteria: **Solvability**: whether the problem is unambiguous and admits a well-defined solution; **Novelty**: rate on a 1–5 scale (higher is better); **Fusion Type**: categorize fusion problems as either conceptual fusion or sequential combination.

**Results.**  The results are summarized below: 1. Solvability: 98.2%. The few unsolvable cases are caused by ambiguous output specifications (e.g., unspecified treatment of multiple valid outputs) rather than logical flaws in the problem statements. 2. Novelty: The mean novelty score is 3.53/5, indicating that the generated problems are reasonably novel. 3. Fusion Type: 77.5% of fused problems are classified as conceptual fusions, indicating deep integration, while 22.5% are classified as sequential combinations, indicating serial concatenation.

*Note: Evaluation based on 113 problems reviewed by expert competitive programmers, with each problem requiring 10–20 minutes of analysis due to complexity.*

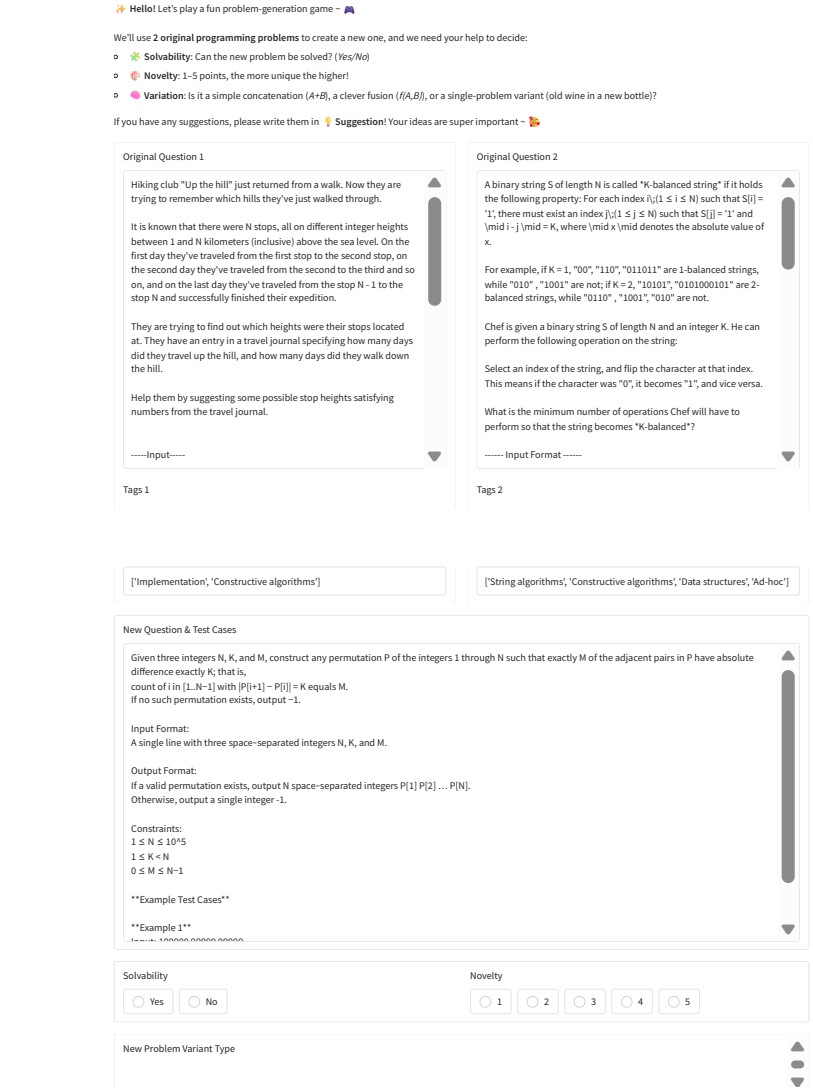

Figure 9: Human rating website.

# G EXAMPLE PROBLEMS

## G.1 SEED PROBLEM 1

**Problem Statement**: Read problems statements in Mandarin Chinese and Russian. Suraj, the Chief Prankster is back in action now and this time he has stolen the valentine's day gift given by Ashi (the love of Chef) to the Chef and ran away with it to Byteland.

Byteland is a not a regular place like Chef's town. The safest way from Chef's town to Byteland is through the path of tasty dishes. The path is named so because there are magical tasty dishes which appear to the traveler that no one can resist eating. Also, Suraj has added a strong sleep potion to each of the dish on this path to stop anyone from following him.

Knowing the devilish nature of Suraj, Ashi is concerned about the Chef and has asked all of Chef's town people to help. The distance from Chef's town to Byteland through the the path of tasty dishes is $X$ units. They have the location where the magic dishes are and how many people are required to eat it completely. Anyone who eats a dish would go to a long sleep and won't be able to continue. They have the information about the tribal clans that live along the the path of tasty dishes who can be of real help in this journey.

The journey Chef and his friends can be described as follows: There is a total of $B$ dishes on the path of tasty dishes. Each dish is located at some distance from Chef's town denoted by $x_i$ for the $i^{th}$ dish ($x_{i-1} < x_i$). To minimize the number of friends Chef has to leave behind, all of them have decided that exactly $y_i$ of them will eat the $i^{th}$ dish, which is the required number of people needed to finish it completely. Also, there are a total of $C$ tribal chef clans, each with their own population and location on the path that Chef and his friends will meet on their way to Byteland. They know that for some clan (say $i$), they are located at a distance of $p_i$ ($p_{i-1} < p_i$) from Chef's town with a population of $r_i$. And if a group of at least $q_i$ men approaches them, they would be able to convince them to join their forces against Suraj.

Given the information about all this, help the Chef to find out the minimum size of the group (including him and his friends) he should start with to reach Byteland and get back Ashi's gift from Suraj.

**Input Format**:

```
The first line of the input contains an integer T denoting the number of test
↪   cases.
Each test case contains three lines which are as follows:
First line of each test case contains X, the distance of Byteland from Chef's
↪   town.
Next line contains an integer B, the number of dishes on the path of tasty
↪   dishes.
Then follows B pairs of space separated integers of the form x_i y_i, where x_i
↪   y_i are as defined above for the i-th dish.
Next line contains an integer C, followed C space separated triplets of integers
↪   p_i q_i r_i as defined above.
```

**Output Format**:

```
For each test case, print the minimum size of the group (including Chef) that is
↪   needed to reach Byteland.
```

**Constraints**:

- $1 \le T \le 10$
- $1 \le X \le 10^9$
- $1 \le B \le 10000$
- Subproblem 1 (25 points): $C = 0$
- Subproblem 2 (75 points): $1 \le C \le 10000$

- $1 \leq x_i < X, x_i < x_{i+1}$
- $1 \leq p_i < X, p_i < p_{i+1}$
- $1 \leq y_i \leq 10^{14}$
- $1 \leq q_i \leq 10^{14}$
- $1 \leq r_i \leq 10^{14}$
- All the positions, of the tasty dishes and tribal clans are distinct.

### G.2 SEED PROBLEM 2

**Problem Statement**: Read problems statements in Mandarin Chinese and Russian.

The Baratheons have been ruling in the Seven Kingdoms for many years. They have seen a lot: prosperity and hunger, tranquility and rebellions, live and death. But nevertheless, they still hold the throne. King Joffrey Baratheon's reign is running now. As a wise man, he honors the history of his family. So, he commanded to build up two monuments, that will remind about some historical periods of the Baratheons. Formally, the Baratheons have been ruling for $N$ years. Every year is described by an integer $A_i$, the level of prosperity in $i$-th year. If $i$-th year was a great year, then $A_i$ might be a positive integer. Otherwise, if $i$-th year was a horrible year, then $A_i$ might be a negative integer. Each historical period can be described as two integers $S$ and $F$, the start and the finish of the period respectively. Of course, $S$ is not greater than $F$ for each period, that we consider in this task. You are to pick two historical periods, but there are some rules:

- Two periods shouldn't have common years. I.e. a period [1, 5] has no common years with a period [6, 7];
- The first period should start earlier than the second one. I.e. a period [1, 5] starts earlier than [6, 7];
- Two periods shouldn't be too close to each other. There must be at least $K$ years between the finish of the first period and the start of the second period. I.e. periods [1, 5] and [10, 10] can be chosen in case $K$ equals to 4, while they can't in case $K$ equals to 5.

The sum of the levels of prosperity in chosen years should be as big as possible. Afterwards, you should report the sum of the levels of prosperity in chosen years to your King. Make everything right, otherwise King Joffrey won't be merciful!

**Input Format**:

```
The first line of the input contains an integer T denoting the number of test
↵   cases.
The description of T test cases follows.
The first line of each test case contains two integers N and K denoting the
↵   length of the Baratheons' reign and the minimal amount of years between two
↵   chosen periods.
The second line contains N space-separated integers A_1, A_2, ..., A_N denoting
↵   the levels of prosperity in corresponding years.
```

**Output Format**:

```
For each test case, output a single line containing the required integer.
```

**Constraints**:

- $1 \leq T \leq 5$
- $2 \leq N \leq 10^5$
- $0 \leq K \leq 10^5$
- $-10^9 \leq A_i \leq 10^9$ for every $1 \leq i \leq N$
- $K + 2 \leq N$

### G.3 NEW PROBLEM

**Problem Statement**: Chef must journey in two phases. **Phase 1: Recruitment** - There are $N$ districts in Chef's town, labeled 1...N. District $i$ has $A_i$ potential volunteers ($A_i$ may be negative: a negative value means the district actually shuns the effort). - Chef may conduct exactly two recruitment campaigns, each on a contiguous interval of districts $[L, R]$. These two intervals must not overlap, and there must be at least $K$ districts between the end of the first and the start of the second. Chef gathers the sum of $A_i$ in each chosen interval. His total recruits $H$ is the sum over both intervals (if that sum is negative, he would of course choose intervals giving non-negative sum). **Phase 2: Expedition** - The path from Chef's town to Byteland has $B$ magical "dishes" at strictly increasing distances $x_i$. To cross dish $i$, exactly $y_i$ members of Chef's party must stop (and thus be lost to sleep). - There are $C$ tribal clans at strictly increasing distances $p_j$. Clan $j$ will join Chef's party and contribute $r_j$ people, but only if at the moment Chef arrives at $p_j$ his current party size is at least $q_j$.

Chef starts Phase 2 with $G_0 + H$ people, where $G_0$ is the size he sets aside before recruitment. As he moves in increasing order of position he encounters dishes and clans. He must ensure that at every dish he has $\geq y_i$ people (to send them to sleep) and that after subtracting $y_i$, his party remains $> 0$. Similarly, at each clan he gains $r_j$ if current $\geq q_j$.

Compute the minimal $G_0$ such that Chef can complete Phase 2 alive.

**Input Format**:

```
T
For each test case:
  N K
  A_1 A_2 ... A_N
  B
  x_1 y_1
  ...
  x_B y_B
  C
  p_1 q_1 r_1
  ...
  p_C q_C r_C
```

**Output Format**:

```
For each test case, print one integer: the minimum G_0.
```

**Constraints**:

- $1 \leq T \leq 5$
- $1 \leq N, B, C \leq 10^5$
- $0 \leq K < N$
- $-10^9 \leq A_i \leq 10^9$
- $1 \leq x_i < x_{i+1} \leq 10^9$
- $1 \leq y_i \leq 10^{14}$
- $1 \leq p_j < p_{j+1} \leq 10^9$
- $1 \leq q_j, r_j \leq 10^{14}$

### G.4 THE LLM-UNSOLVABLE PROBLEM

**Problem Statement**: You have $2n$ heroes, the $i$-th hero has initial power $p_i$. There are $m$ special pairs, each pair is two heroes; a special pair must be assigned to the same team. You must split all $2n$ heroes into two teams of size $n$: your team and the other team. After the split, you can perform any number of operations on each hero on your team: in one operation you choose a hero and add $x$

to its power or subtract $x$ from its power (but hero's power cannot go below 0). Finally, define the MEX of your team's powers as the smallest non-negative integer not present in the multiset of final powers. Compute the maximum MEX you can achieve by choosing an optimal split.

**Input Format**:

- Line 1: three integers $n, m, x$ ($1 \le n \le 2 \cdot 10^5, 0 \le m \le n, 1 \le x \le 10^5$)
- Line 2: $2n$ integers $p_1, p_2, \ldots, p_{2n}$ ($0 \le p_i \le 10^9$)
- Next $m$ lines: two integers $a, b$ ($1 \le a < b \le 2n$), denoting a special pair (heroes $a$ and $b$); pairs are disjoint.

**Output Format**: One integer—the maximum MEX you can obtain on your team under the best assignment and any number of $\pm x$ operations.

**Constraints**:

- Special pairs are disjoint; each hero belongs to at most one pair.
- You must put exactly $n$ heroes on "your team."
- If heroes $a$ and $b$ form a special pair, both must go either to your team or to the other team.

## H INTEGRATED PROMPTS FOR TEST CASE GENERATION

```
1  I will provide you with a programming problem description, and your
        task is to generate standardized test input samples using the
        CYaRon library.
2
3  You need to complete the following steps:
4  1. Parse the constraints on the input from the problem description,
        such as the range of input data, specific input constraints, etc.
5  2. Write a function generate_test_input using the CYaRon library to
        randomly generate test inputs based on a specified problem size.
        The function should validate that the parameters fall within the
        specified constraints. If any parameter is out of range, the
        function should return None. If the parameters are valid, generate
        a random test input and return an input string (input_string).
6  3. Write a function validate_test_input to verify whether the
        generated test input satisfies the requirements specified in the
        problem description. This includes checking the input data type
        and constraints parsed in step 1, such as range and other
        conditions. The function should take input_string as input and
        return a boolean (True/False).
7
8  Output format (strictly follow)
9  Part 1: Parse Input Constraints
10 Specify the input constraints as described in the problem.
11
12 Part 2: Code for Test Input Generation
13 import cyaron as cy #cyaron version: 0.7.0
14
15 def generate_test_input():
16     # set parameters constraints that meet requirements (e.g. 1 <= N
            <= 300)
17     ...
18     # Generate input using CYaRon
19     input_data = [
20         ...
21     ]
22     return "\n".join(map(str, input_data))
23
24 Part 3: Code to Validate Test Input
25 def validate_test_input(input_string):
26     # Validation logic
27     return <boolean>
28
29 Note:
30 - cy.Integer() is not supported; it should be cy.randint.
31 - use cy.String.random instead of cy.String
32 - The function generate_test_input() should not accept any parameters.
        You need to generate the input entirely within the function.
33 - Generate code following the above format, without starting with ```
        python or similar markers.
```

Listing 1: Random Input Generator (CYaRon)

```
1  I will provide you with a programming problem description, and your
        task is to generate adversarial test input samples using the
        CYaRon library.
2
3  You need to complete the following steps:
4  1. Parse the constraints on the input from the problem description,
        such as the range of input data, specific input constraints, etc.
5  2. Write a function generate_test_input using the CYaRon library to
        generate a single adversarial test input designed to challenge
        boundary conditions or worst-case complexity. The function should
```

```
      internal1y randomize which adversarial strategy to use, without
      accepting any parameters. The generated input should still conform
       to the problem constraints.
3. Write a function validate_test_input to verify whether the
      generated test input satisfies the requirements specified in the
      problem description. This includes checking the input data type
      and constraints parsed in step 1. The function should take
      input_string as input and return a boolean (True/False).

Output format (strictly follow):
Part 1: Parse Input Constraints
Specify the input constraints as described in the problem.

Part 2: Code for Test Input Generation
import cyaron as cy # cyaron version: 0.7.0
import random

def generate_test_input():
    # set parameters constraints that meet requirements (e.g. 1 <= N
        <= 300)
    ...
    # Randomly choose one adversarial strategy
    strategy = random.choice(["equal_weights", "
        alternating_large_small", "large_ends"])

    if strategy == "equal_weights":
        # example: all weights are maximal
        N = cy.randint(100000, 100000) # fix to worst-case size
        max_weight = cy.randint(109, 109)
        k = random.choice([1, N//2, N-1, N])
        weights = [max_weight] * N
    elif strategy == "alternating_large_small":
        N = cy.randint(100000, 100000)
        max_weight = cy.randint(109, 109)
        k = random.choice([1, N//2, N-1, N])
        weights = [max_weight if i%2 else 1 for i in range(N)]
    else: # large_ends
        N = cy.randint(100000, 100000)
        max_weight = cy.randint(109, 109)
        k = random.choice([1, N//2, N-1, N])
        weights = [max_weight] + [1]*(N-2) + [max_weight]

    # Build input string
    input_lines = [f"{N} {k}"] + [str(w) for w in weights]
    return "\n".join(input_lines)

Part 3: Code to Validate Test Input
def validate_test_input(input_string):
    try:
        lines = input_string.strip().split('\n')
        N_k = lines[0].split()
        if len(N_k) != 2:
            return False
        N, k = map(int, N_k)
        if not (1 <= N <= 100000):
            return False
        if not (1 <= k <= N):
            return False
        weights = list(map(int, lines[1:]))
        if len(weights) != N:
            return False
        for w in weights:
            if not (1 <= w <= 109):
                return False
        return True
```

```
62      except:
63          return False
64
65  Note:
66  - generate_test_input() must return a single adversarial input string,
        not a list.
67  - Use cy.randint instead of cy.Integer().
68  - The function should generate adversarial yet valid data fully inside
        , without parameters.
```

Listing 2: Adversarial Input Generator (CYaRon)

```
1  Task:
2   Generate a challenging test input for the algorithm problem:
3   {problem_description}
4   Instructions:
5   - Focus on edge cases or scenarios that maximize the failure
        probability in faulty solutions.
6   - Due to the output length limit, you should generate a small-scale
        test input that is complete and valid.
7   - Output the test input directly, not code to generate it.
8   Output format:
9   '''plaintext
10  {test input}
11  '''
12  Think step by step.
```

Listing 3: Direct Test Input Generator

# I INTEGRATED PROMPTS FOR ALGORITHMIC PROBLEM GENERATION

```
1  You are an expert competitive programmer.
2  I'll provide you with two programming problems.
3  1. Explore how to combine these concepts
4  2. Design a new challenging problem that integrates them
5
6  Output format(strictly follow):
7   ## Part 1: Original Problems and Solution Analysis
8   Step1: [Describe the steps of reasoning]
9   Step2: xxx
10  ...
11
12  ## Part 2: New Problem Description:
13  New_problem: [Describe the new problem clearly in natural language.]
14
15  Input Format: [Specify the input format]
16  Output Format: [Specify the output format]
17
18  ## Part 3: Example Test Cases
19  Input: [Input for test case 1]
20  Output: [Expected output for test case 1]
21  Input: [Input for test case 2]
22  Output: [Expected output for test case 2]
23
24  ## Part 4: Category
25  difficulty: [Easy/Medium/Hard]
26  tags: [tags of new problem, separated by commas.]
27  skills: [skills of new problem, separated by commas.]
28
29  Note:
30  1. select the 1-3 most relevant skills from the given list, ranked by
        relevance.
```

```
31  2. The new problem must be rigorous and clearly stated, and include
        explicit input/output specifications or constraints.
32  3. Please design questions that have one correct answer; avoid 'output
         one possible combination' that could have multiple valid answers.
33  4. Provide two example test cases to demonstrate the new problem.
```

Listing 4: Prompts for Cross-type Fusion

```
1   You are an expert competitive programmer.
2   I'll provide you with one programming problem, its solution, and the
        key concepts they test.
3   You need to:
4   1. Analyze its problem design approaches
5   2. Create a new variation question based on the original one
6
7   Output format(strictly follow):
8    ## Part 1: Original Problems and Solution Analysis
9    Step1: [Describe the steps of reasoning]
10   Step2: xxx
11   ...
12
13   ## Part 2: New Problem Description:
14   New_problem: [Describe the new problem clearly in natural language.]
15
16   Input Format: [Specify the input format]
17   Output Format: [Specify the output format]
18
19   ## Part 3: Example Test Cases
20   Input: [Input for test case 1]
21   Output: [Expected output for test case 1]
22   Input: [Input for test case 2]
23   Output: [Expected output for test case 2]
24
25   ## Part 4: Category
26   difficulty: [Easy/Medium/Hard]
27   tags: [tags of new problem, separated by commas, referring to the
         tags of the original problems.]
28   skills: [skills of new problem, separated by commas.]
29
30  Note:
31  1. select the 1-3 most relevant skills from the given list, ranked by
         relevance.
32  2. The new problem must be rigorous and clearly stated, and include
         explicit input/output specifications or constraints.
33  3. Provide two example test cases to demonstrate the new problem.
```

Listing 5: Prompts for Single-problem Extension

