# OpenReview forum: "UniCode: A Framework for Generating High-Quality Competitive Coding Problems"
_ICLR.cc/2026/Conference — Submitted to ICLR 2026_

### Official Review · Reviewer_xybC · 2025-10-27

**Soundness:** 3
**Presentation:** 3
**Contribution:** 2
**Rating:** 6
**Confidence:** 4

**Summary:**

The paper presents UniCode, a framework for automatically generating competitive programming problems and test cases to reduce data contamination and improve scalability of code benchmarks. It expands seed problems through single-problem extension, same-type fusion and cross-type fusion, and builds reliable test sets via a stress-driven pipeline. The resulting dataset includes 492 verified problems.

Experiments on 19 code models show strong discriminative power: even the best model reaches only 70.3% pass@1 and drops notably on adversarial cases. Models generalize well to paraphrased tasks but fail on structurally novel ones, suggesting limited algorithmic generalization. UniCode thus offers a scalable way to evaluate reasoning in code LLMs.

**Strengths:**

1. A well-designed, end-to-end test generation pipeline that avoids reference solutions. The per–test case majority voting between brute-force and optimized LLM solvers produces reliable labels while still preserving challenging instances.

2. Practical dataset construction with clear procedures for input synthesis, difficulty calibration, problem taxonomy, and explicit time/memory budgets per test, making the benchmark genuinely usable rather than merely conceptual.

3. A strong stance on contamination and robustness: the evolutionary problem generation (single-problem extension, intra-/cross-type fusion) yields structurally novel problems beyond paraphrases, mitigating leakage and probing true algorithmic generalization.

4. High discriminative power across 19 models with transparent reporting (rand vs. adv), clearly exposing brittleness to adversarial inputs and the gap between linguistic robustness and algorithmic generalization.

**Weaknesses:**

1. Limited ablations: while the pipeline is carefully engineered, the necessity of each stage is not well justified. The paper does not probe key choices such as per–test case vs. per–problem voting, solver diversity, or the 20/20/10 suite composition; adding an “LLM-pass” alongside Rand/Adv would also clarify the marginal value of LLM-synthesized inputs.

2. Quality assurance remains narrow: per-testcase voting can still admit residual mislabeled cases, and the human validation sample is small

**Questions:**

Regarding the 20/20/10 suite composition, beyond the exact ratios, could you justify the necessity of including G_llm? Please add a G_llm-Pass metric (analogous to RandPass/AdvPass) to quantify its marginal value, and include a small sensitivity check showing results with and without G_llm inputs.

---

> ### Author Response · Authors · 2025-11-19
>
> Thank you for your careful review. I will answer your questions point by point below.
> > ​Q1. Limited ablations: while the pipeline is carefully engineered, the necessity of each stage is not well justified. The paper does not probe key choices such as per–​**​**test case**​**​ vs. per–problem voting, solver diversity, or the 20/20/10 suite composition
>
> We appreciate the reviewer’s concern regarding ablations on per–test case vs. per–problem voting, solver diversity, and the 20/20/10 suite composition. We provide further clarifications and additional experimental results below.
>
> **​(a) Per–​**​**test case**​**​ vs. per–problem voting**
>
> As discussed in §3.3, per–test case voting offers the following advantages:
>
> * **Preserving valuable hard problems.** per–problem voting aggregates at the *solution level* and discards a problem unless a majority of solutions agree on *all* test cases. Our aggregation is performed *per ​*​​*test case*​, so disagreement on a few cases does not invalidate the entire problem. This enables us to retain substantially more high-quality, diverse problems.  As shown in Table 1, our number of \`\`problems with validated suites” is slightly higher than the baseline, because per–test case voting avoids excessive discarding.
> * **Stronger bug-detection capability.** In practice, a problem’s suite may contain a few particularly hard or boundary test cases where solvers disagree, while most others are consistent. Per–test case voting pinpoints and isolates these difficult cases rather than discarding the whole problem.  Empirically, this approach increases *coverage* from ​80.2% → 86.0%​, and allows us to capture significantly more buggy code in downstream evaluation.
>
> **(b) About solver diversity**
>
> Before selecting the single o4mini as our solver, we conducted a small-scale controlled comparison to assess whether using multiple models as candidate solvers would enhance verification performance compared to relying on a single strong model. We evaluated 3 settings on 52 problems, and adopted the following 3 settings for majority voting:
>
> * Two models (Deepseek-r1, o4-mini): each generates 6 candidate solutions.
> * Three models (Deepseek-r1, o3-mini, GPT-4.1): each generates 4 candidate solutions.
> * Single strong model (o4-mini): generates 12 independent candidate solutions.
>
> Results are shown below. Using the single strongest model (o4-mini) to produce multiple independent solvers gives the best correctness; mixing strong and weaker generators does not improve and can degrade overall performance because weaker generators produce more incorrect candidates that lower the ensemble’s effective signal. Similar findings are also observed in other recent works [1,2]. We therefore recommend using the best available model for multi-solution generation; if cost is a concern, a small ensemble of top-tier reasoning models is an acceptable alternative.
>
> | setting | model                         | correctness     | coverage        |
> | --------- | ------------------------------- | ----------------- | ----------------- |
> | 1       | Deepseek r1 & o4mini          | 92.3%           | 87.3%           |
> | 2       | Deepseek r1 & o3mini & gpt4.1 | 88.5%           | 86.0%           |
> | 3       | o4mini                        | **94.2%** | **87.5%** |
>
> [1] Trad, Fouad, and Ali Chehab. "To ensemble or not: Assessing majority voting strategies for phishing detection with large language models." *International Conference on Intelligent Systems and ​*​​*Pattern Recognition*​. Cham: Springer Nature Switzerland, 2024.
>
> [2] Theisen, Ryan, et al. "When are ensembles really effective?." *Advances in neural information processing systems* 36 (2023): 15015-15026.
>
> **(c) Regarding the 20/20/10 suite composition**
>
> We selected the 20/20/10 composition based on an empirical sweep across multiple distributions, evaluating correctness and coverage over 48 problems and 960 human-crafted solutions. Representative results include:
>
> * (5, 5, 0): correctness 97.9, coverage 77.0
> * (10, 10, 5): correctness 95.8, coverage 81.4
> * (20, 20, 10): correctness 94.0, coverage 87.5 ← selected
> * (30, 30, 20): correctness 91.7, coverage 88.7
>
> From these results, we observed that (a) too few test cases significantly weakens fault detection, (b) too many increases sensitivity to single-case noise and pipeline cost, (c) 20/20/10 balances detection power and robustness while controlling cost. We include these experiments in Appendix D.1

---

> > ### Author Response · Authors · 2025-11-19
> >
> > > Q2. ​**​**Human**​**​ ​**​**validation**​**​ sample is small
> >
> > We conducted an additional round of annotation with 63 new problems, bringing the total to 113 problems (50 initial + 63 new) evaluated by expert competitive programmers. Solvability is 98.2%, the few unsolvable cases were due to ambiguous output specifications (e.g., unspecified handling of multiple valid outputs), not logical flaws in the problem itself. While the sample size may seem modest, each problem requires 10–20 minutes of expert review due to their complexity. Evaluating 113 problems is a substantial effort. We hope that our extended evaluation of generated problems can address your concern.
> >
> > > Q3. per-testcase voting can still admit ​**​**residual**​**​ mislabeled cases
> >
> > We acknowledge that this is a limitation of generative evaluation. Since our goal is to scale up, the pipeline cannot be entirely error-free, and the same issue appears in prior work such as the ICML spotlight paper ​*MCU: An Evaluation Framework for Open-Ended Game Agents*​.
> >
> > However, we make every effort to minimize this error by introducing a three-stage stress-driven pipeline consisting of Brute-Force Validation, Consensus Validation, and LLM Adjudication. This keeps the error rate small, with correctness improving from 86.9 to 94.5. In addition, Appendix B provides mathematical proof that even when a benchmark contains a small fraction of erroneous tasks, the reported accuracy remains trustworthy. Therefore, UniCode achieves a stronger balance: it retains challenging and informative problems while keeping contamination low, which is essential for scaling generative evaluation and faithfully assessing models’ true generalization.

---

> ### Author Response · Authors · 2025-11-19
>
> > Q4. Could you justify the necessity of including G\_llm?  ​Please add a G\_llm-Pass metric (analogous to RandPass/AdvPass) to quantify its marginal value, and include a small sensitivity check showing results with and​ without G\_llm inputs.
>
> Thank you for the question. In the original version, we merged adv and llm-pass results because both represent challenging cases. Now we update Table 2 accordingly: 1. We separate G\_llm Pass from AdvPass. 2. We add remove Adv / remove Rand / remove G\_llm ablation.
>
> | Model                     | Adv Pass  | Rand Pass | G_llm Pass | w/o Adv  | w/o Rand | w/o G_llm | Overall |
> |----------------------------|-----------|-----------|------------|----------|----------|-----------|---------|
> | o4-mini (high)*            | 80.86%    | 80.11%    | 76.74%     | 76.86%   | 72.78%   | 80.50%    | 70.3%   |
> | gpt-5 (medium)*            | 79.02%    | 75.92%    | 74.59%     | 73.33%   | 70.03%   | 76.57%    | 67.7%   |
> | google/gemini-2.5-pro*     | 75.35%    | 74.71%    | 70.47%     | 68.80%   | 64.43%   | 70.94%    | 61.6%   |
> | qwen3-235b-a22b            | 64.04%    | 68.55%    | 66.35%     | 63.35%   | 55.84%   | 66.82%    | 53.5%   |
> | deepseek-chat-v3.1         | 62.54%    | 61.05%    | 57.54%     | 53.85%   | 51.83%   | 57.94%    | 49.8%   |
> | gemini-2.5-flash*          | 59.30%    | 61.82%    | 64.06%     | 55.42%   | 50.29%   | 53.97%    | 47.7%   |
> | grok-3-mini*               | 55.43%    | 61.76%    | 63.81%     | 56.36%   | 48.47%   | 52.77%    | 46.4%   |
> | claude-3.7-sonnet*         | 51.43%    | 63.86%    | 66.29%     | 56.96%   | 44.29%   | 50.66%    | 45.5%   |
> | gpt-4.1-mini*              | 54.52%    | 58.38%    | 56.90%     | 51.23%   | 45.20%   | 49.79%    | 42.4%   |
> | gpt-4.1*                   | 49.33%    | 51.96%    | 52.91%     | 46.72%   | 39.47%   | 46.91%    | 36.5%   |
> | qwen3-coder                | 45.20%    | 53.91%    | 54.65%     | 45.90%   | 36.72%   | 45.27%    | 35.4%   |
> | claude-sonnet-4*           | 41.81%    | 48.32%    | 60.56%     | 45.49%   | 33.90%   | 37.45%    | 32.4%   |
> | llama-4-maverick           | 37.33%    | 41.34%    | 43.39%     | 35.66%   | 27.47%   | 34.98%    | 26.2%   |
> | gpt-4o*                    | 22.88%    | 24.02%    | 22.54%     | 17.62%   | 14.97%   | 21.81%    | 15.4%   |
> | qwen-2.5-32b-coder         | 20.34%    | 26.82%    | 18.59%     | 13.11%   | 11.58%   | 18.93%    | 13.4%   |
> | gemma-3-27b-it             | 17.68%    | 21.22%    | 37.53%     | 17.37%   | 13.26%   | 16.60%    | 13.1%   |
> | llama-3.3-8b-instruct      | 9.60%     | 12.85%    | 12.70%     | 5.33%    | 5.87%    | 5.76%     | 5.5%    |
> | **Average Pass Rate**       | **52.99%**    | **55.92%**    | **55.53%**     | **49.94%**   | **44.53%**   | **51.21%**    | **38.99%**  |
>
> As shown above, Rand cases are easier and contribute most of the base correctness of models. Adv and LLM-syn are effective stress tests. Removing Adv or G\_llm increases the pass rate (49.94% and 51.21%), indicating these sets are significantly harder. Therefore, G\_llm adds meaningful incremental difficulty and stress-test coverage. The mixed design ensures comprehensive evaluation across both common and edge-case scenarios. We include these experiments in Appendix D.2.

---

### Official Review · Reviewer_2L99 · 2025-10-29

**Soundness:** 3
**Presentation:** 3
**Contribution:** 2
**Rating:** 4
**Confidence:** 3

**Summary:**

The paper presents UniCode, a generative evaluation framework for creating novel competitive-programming problems with corresponding test suites, aiming to reduce contamination and improve scalability over static benchmarks. It diversifies tasks via three strategies—extension, same-type fusion, and cross-type fusion—and constructs test suites without canonical solutions through stress-testing, majority voting, and LLM adjudication. The resulting dataset includes 492 problems across 15 tags, evaluated on 19 LLMs, with the best achieving ~70% pass@1. Human evaluation reports 98% solvability, and ablations show improved correctness (94.5%) and coverage (86.0%) over baseline methods.

**Strengths:**

- It addresses contamination and saturation issues in code benchmarks by proposing a generative evaluation framework that automatically produces new, diverse programming tasks.
- The multi-stage test-suite construction (stress-testing, filtering, majority voting, adjudication) effectively removes the need for canonical solutions and yields high correctness and coverage.
- The benchmark demonstrates strong discriminative power across 19 models, supported by human validation and ablation analyses indicating robustness and scalability.

**Weaknesses:**

- The paper does not discuss or compare with Evol-Instruct and its derivative WizardCoder, which pioneered LLM-based progressive task evolution. The proposed single-problem extension is conceptually similar to Evol-Instruct’s “in-depth evolution,” undermining the novelty of the generation component.
- The entire framework relies heavily on a single closed-source model (o4-mini) for both task generation and final adjudication. This creates potential “generator bias,” as the benchmark might inadvertently favor models with similar capabilities or architectures.
- The framework depends on seed problems from TACO, yet does not analyze whether these seeds or their variants exist in common pretraining corpora. Without such an overlap check, the contamination-resistance claim remains unverified.

**Questions:**

- Why are Evol-Instruct and WizardCoder not cited or compared? How is your single-problem extension fundamentally different from their progressive-evolution mechanism?
- Have you tested alternative (preferably open-source) generators or adjudicators? How does this affect solvability rates or model rankings?
- What are the dominant sources of the 5.5% incorrect test cases—faulty brute-force baselines, consensus failures, or adjudication errors?
- Does “cross-type fusion” always yield truly hybrid algorithmic problems, or are they often decomposable into sequential subproblems from the seed set?

---

> ### Author Response · Authors · 2025-11-19
>
> Thank you for your reply. We will address your questions point by point below.
> > Q1. Why not cite/compare Evol-Instruct / WizardCoder? Is single-problem extension different?
>
> Thank you for the reminder. We have added explicit citations and a brief comparative paragraph to ​*Related Work*​. Although the ideas may appear related, our motivation and methodology differ fundamentally.
>
> * **WizardCoder / Evol-Instruct** focus primarily on *instruction complexity* and *curriculum-style or instruction-evolution methods* (e.g., making prompts harder or more varied), enabling models to learn to handle more complex instructions during training.
> * **UniCode**​**​ ​**targets algorithmic problem generation: we generate genuinely new algorithmic challenges by altering the underlying algorithmic core rather than simply making the problem description more complex. For example, *Two Sum → Three Sum* changes the required solution class (two-sum hash → three-sum two-pointer or sorting variants), demanding different algorithmic reasoning rather than a harder instruction.
>
> Therefore, **unlike text-based transformation strategies in WizardCoder (e.g., adding 10 extra words, or replacing a common requirement with a less common one),** which tend to produce questions that are linguistically more complex but not fundamentally different in algorithmic (as shown in Figure 3, where ShadowQS and SeedQS yield similar performance), UniCode focuses on synthesizing truly new problems through novel formulations or new combinations of algorithmic concepts. In addition, while Evol-Instruct and WizardCoder primarily focus on generating richer data to train models, our work tackles another core challenge: producing test cases ​**without access to ground-truth solutions**​, which pose a significant difficulty for building a reliable benchmark.
>
> > Q2. Heavy reliance on o4-mini for generation/adjudication (generator/adjudicator bias)? Have you tested alternative (preferably open-source) generators or adjudicators? How does this affect solvability rates or model rankings?
>
> We are aware of this potential concern and explicitly stated it as a limitation in the conclusion “A limitation of our study is its reliance on a single powerful closed-source LLM, o4-mini, which may introduce bias by generating problems that align with its own strengths”.
>
> To examine this effect, we additionally used the open-source model DeepSeek-R1 to generate 104 problems across five tags and evaluated multiple models with different capability levels. Due to time constraints, we manually verified the solvability of a small to medium-sized sample of 25 problems and found no unsolvable cases. The problems were well-formulated with clear logic. The results show no clear evidence that a model performs substantially better on problems it generated itself. **Compared with human-designed problems and those generated by o4-mini, the ranking of ​**​**LLM**​**​ performance remains consistent. ​**Statistically, the Unicode benchmarks produced by different generators are highly aligned **Pearson r = 0.984, p = 4e-4** (Unicode from deepseek vs. Unicode from o4-mini).
>
> | model                     | Unicode (deepseek) | Unicode (o4mini) | LiveCodeBench (human, no contamination) |
> | --------------------------- | ------------------------------ | ---------------------------- | ----------------------------------------- |
> | gpt-5-2025-08-07          | 0.725                        | 0.677                      | —                                      |
> | o4-mini-2025-04-16        | 0.702                        | 0.661                      | 0.742                                   |
> | deepseek-r1-0528          | 0.616                        | 0.556                      | 0.731                                   |
> | o3-mini-2025-01-31        | 0.51                         | 0.551                      | 0.63                                    |
> | gpt-4.1-mini-2025-04-14   | 0.413                        | 0.424                      | 0.532                                   |
> | google/gemma-3-27bit:free | 0.146                        | 0.131                      | —                                      |
>
> Different generators may favor slightly different difficulty distributions, which affects absolute scores to some degree, but does not materially change model ranking or the main conclusions. Therefore, to further mitigate generator-source risk, we will release **Unicode-Multi ​**after the rebuttal period. It is a consolidated benchmark co-created by multiple state-of-the-art LLMs (including o1-mini, GPT-4o, Gemini-2.0-Pro, DeepSeek-R1, and Qwen-32B). Each model contributed approximately 100 problems to ensure greater diversity. We have added a detailed analysis of generator bias in Section 4.5.

---

> ### Author Response · Authors · 2025-11-19
>
> >Q3. Have you tested alternative adjudicators?
>
> Yes, our approach to adjudicator selection was determined through a controlled experiment. We understand that your concern is not limited to the adjudicator, which is only a small component and contributes only marginally to performance differences (refer to Q4). The key question is whether using a single o4-mini model throughout test case synthesis could introduce bias, and whether combining multiple models with majority voting might lead to better verification performance.
>
> To address this, We ran a controlled comparison to test whether combining multiple models as candidate generators improves verification performance, compared to using only a single strong model. We evaluated 3 settings on 52 problems, and adopted the following 3 settings for majority voting:
>
> * Two models (Deepseek-r1, o4-mini): each generates 6 candidate solutions.
> * Three models (Deepseek-r1, o3-mini, GPT-4.1): each generates 4 candidate solutions.
> * Single strong model (o4-mini): generates 12 independent candidate solutions.
>
> Results are shown below. Using the single strongest model (o4-mini) to produce multiple independent solvers gives the best correctness; mixing strong and weaker generators does not improve and can degrade overall performance because weaker generators produce more incorrect candidates that lower the ensemble’s effective signal. Similar findings are also observed in other recent works [1,2]. We therefore recommend using the best available model for multi-solution generation; if cost is a concern, a small ensemble of top-tier reasoning models is an acceptable alternative.
>
> | setting | model                         | correctness     | coverage        |
> | --------- | ------------------------------- | ----------------- | ----------------- |
> | 1       | Deepseek r1 & o4mini          | 92.3%           | 87.3%           |
> | 2       | Deepseek r1 & o3mini & gpt4.1 | 88.5%           | 86.0%           |
> | 3       | o4mini                        | **94.2%** | **87.5%** |
>
> [1] Trad, Fouad, and Ali Chehab. "To ensemble or not: Assessing majority voting strategies for phishing detection with large language models." *International Conference on Intelligent Systems and ​*​​*Pattern Recognition*​. Cham: Springer Nature Switzerland, 2024.
>
> [2] Theisen, Ryan, et al. "When are ensembles really effective?." *Advances in neural information processing systems* 36 (2023): 15015-15026.
>
> > Q4. What are the dominant sources of the 5.5% incorrect test cases—faulty brute-force baselines, consensus failures, or adjudication errors?
>
> We carried out an error analysis and found that:
>
> * Consensus failures (\~3.1%) : multiple solvers agree on an identical wrong output for large inputs.
> * Brute-force baseline faults (\~1.4%) ： rare cases where multiple brute-force variants share a bug on small inputs that propagates.
> * LLM adjudication errors (\~1.0%) : the adjudicating LLM occasionally misjudges among the top-two outputs.
>
> > Q5. Are cross-type fusions genuinely hybrid or just sequential composition?
>
> Based on an evaluation of 113 problems by expert competitive programmers, we found that Conceptual fusion accounts for 77.5%, while series combination accounts for 22.5%. This indicates that the majority of fusions are not simple concatenations, but rather involve skillful conceptual integration.
>
> > Q6. The framework depends on seed problems from TACO, yet does not analyze whether these seeds or their variants exist in common pretraining corpora. Without such an overlap check, the contamination-resistance claim remains unverified.
>
> Thank you for your question.**​ ​**Our approach does not suffer from this concern, because **even if seed problems are contaminated, ​**UniCode’s generative nature allows for continuous creation of new problems **that models cannot easily memorize. ​**Specifically, as demonstrated in Line399-401: we constructed three problem sets derived from 50 problems in​**​ LiveCodeBench v1.2**​, an *early version* that carries **potential data-contamination risk** (the leaderboard marks contamination risk in red https://livecodebench.github.io/leaderboard.html). We selected models that are marked as contaminated claude-3.5 and qwen3-235b-a22b, and used our UniCode generation procedure to synthesize new problem variants from these contaminated seeds.
>
> From Figure 3, we observed a **substantial performance drop on UniCode-generated problems versus the original seeds** (​**Claude: 0.70 → 0.21; Qwen3: 0.89 → 0.58**​). As you mentioned, “If LLM's performance is significantly lower on the generated problems, it will demonstrate the consequence of data contamination and further motivates the proposed approach.” Thus, the large gap between seeds and their UniCode variants shows that UniCode produces truly novel problems and offers a stronger test of LLMs' generalization.
>
> We have revised Section 4.3 to provide a clearer interpretation of the contamination analysis.

---

### Official Review · Reviewer_bwy7 · 2025-10-30

**Soundness:** 3
**Presentation:** 3
**Contribution:** 3
**Rating:** 4
**Confidence:** 4

**Summary:**

The paper is devoted to introduction of a novel code generation benchmark. The methodology includes a step of verification via small input brute-force solver.

Personally, I like the idea of creating such a benchmark, although I do not see how it solves an issue of "vast memorization of training data", since this benchmark in 6 months will be in training dataset for every new model.

Another important issue - it is unclear how small input solver helps with big input, since in programming competitions it is usually the case, that some algorithm is not able to work with bigger input for the same problem. The authors report 98% of solvability of generated problems, although they evaluated only 50 problems, that means that one problem gives 2% error.

In closing, I would like to say that the proposed methodology seems to be interesting, but the benchmark itself does not solve an issue of previously introduced benchmarks.

**Strengths:**

novel methodology to generate problems

**Weaknesses:**

Personally, I like the idea of creating such a benchmark, although I do not see how it solves an issue of "vast memorization of training data", since this benchmark in 6 months will be in training dataset for every new model.

Another important issue - it is unclear how small input solver helps with big input, since in programming competitions it is usually the case, that some algorithm is not able to work with bigger input for the same problem. The authors report 98% of solvability of generated problems, although they evaluated only 50 problems, that means that one problem gives 2% error.

**Questions:**

a) Please describe how your benchmark solves a mentioned issue?

b) Please evaluate the generated problems more thoroughly.

---

> ### Author Response · Authors · 2025-11-19
>
> Thank you for your valuable feedback, We will address your concern as follows.
>
> > Q1. Personally, I like the idea of creating such a benchmark, although I do not see how it solves an issue of "vast memorization of ​**​​**training data**​**", since this benchmark in 6 months will be in training dataset for every new model.
>
> Thank you for your interest in our idea. In fact, we have already demonstrated in Section 4.3 of our paper that our new method effectively addresses the "vast memorization of training data" issue.
>
> **As shown in Section 4.3, even if seed problems are contaminated, our method produces novel variants that models cannot easily memorize. ​**Specifically, we constructed three problem sets derived from ​**50 problems in LiveCodeBench v1.2**​, an *early version* that carries **potential data-contamination risk** (the leaderboard marks contamination risk in red https://livecodebench.github.io/leaderboard.html). We selected two models that are marked as contaminated: claude-3.5 and qwen3-235b-a22b, **and used our ​**​**UniCode**​**​ generation procedure to synthesize new problem variants from these contaminated seeds.**
>
> From Figure 3, we observed a **substantial performance drop on UniCode-generated problems versus the original seeds** (​**Claude: 0.70 → 0.21; Qwen3: 0.89 → 0.58**​). To be explicit, the contamination concern is that some test problems (or very similar variants) appear in model training data, so high scores on those seeds may reflect memorization rather than generalization. To evaluate this, UniCode produces new problems that preserve the same knowledge points but restate and probe them in novel ways (e.g., new formulations or different compositions of subproblems). The large gap between the seed problems and their UniCode variants shows that UniCode produces truly novel problems and provides a stronger test of LLMs' generalization.
>
> We have revised Section 4.3 to provide a clearer interpretation of the contamination analysis.
>
> > Q2. it is unclear how small input solver helps with big input, since in programming competitions it is usually the case, that some algorithm is not able to work with bigger input for the same problem. ​
>
> Thank you for your question. We would like to clarify that **the small-input solver in ​**​**UniCode**​​**​ is not intended to directly solve large-scale problems**​. Instead, its purpose is to establish a **trusted verification baseline** that supports our multi-stage test synthesis pipeline. Here’s how it works:
>
> * We use brute-force solvers to generate reliable outputs for small-scale inputs (e.g., \\( n \\leq 10^4 \\)), where exhaustive computation is feasible.
> * These trusted outputs are used to stress-test and filter a pool of LLM-generated optimized solutions. Only those that pass all small-input tests are admitted into the trusted solver pool.
> * For large-scale inputs (e.g., \\( n \\approx 2 \\times 10^7 \\)), where brute-force is infeasible, we rely on majority consensus among the optimized solvers from the trusted pool.
>
> **This approach ensures that the optimized solvers used for large-input ​**​**validation**​​**​ have already been verified for correctness on simpler cases**​, thereby improving the overall reliability of the test cases without assuming that small-input algorithms scale directly.
>
> > ​Q3. The authors report 98% of solvability of generated problems, although they evaluated only 50 problems**​**, that**​**​ means that one problem gives 2% error. Please evaluate the generated problems more thoroughly.
>
> We conducted an additional round of annotation with 63 new problems, bringing the total to 113 problems (50 initial + 63 new) evaluated by expert competitive programmers. Each annotator was provided with two original seed problems along with the newly generated ones, and was asked to evaluate the new problems based on the following criteria: 1. Solvability: Whether the problem is unambiguous and admits a well-defined solution. 2. Novelty: Rated on a 1–5 scale (higher is better).3. Fusion Type: Categorized as either *conceptual fusion* or ​*sequential combination*​. The results are listed as follows:
>
> * Solvability: 98.2% – the few unsolvable cases were due to ambiguous output specifications (e.g., unspecified handling of multiple valid outputs), not logical flaws in the problem itself.
> * Novelty: Average score of 3.53/5, indicating that experts found the problems to be reasonably novel.
> * Fusion Quality: 77.5% of fused problems were rated as *conceptual fusions* (deeply integrated), while only 22.5% were *sequential combinations* (serial concatenation).
>
> While the sample size may seem modest, each problem requires 10–20 minutes of competitive programmers review due to their complexity. Evaluating 113 problems is a substantial effort. We hope that our extended evaluation of the quality and novelty of the generated problems addresses your concern.
>
> We have included these results in Appendix F.

---

### Official Review · Reviewer_tfss · 2025-10-31

**Soundness:** 2
**Presentation:** 3
**Contribution:** 3
**Rating:** 4
**Confidence:** 4

**Summary:**

This paper presents UniCode, a novel framework for automatically generating high-quality and contamination-resistant algorithmic coding problems and test cases to overcome the limitations of static, human-authored benchmarks. Drawing inspiration from biological evolution, UniCode leverages Large Language Models (LLMs) to diversify problems through single-problem extension, same-type fusion, and cross-type fusion, producing more varied and challenging tasks. Using this framework, the authors curate a benchmark of 492 algorithmic problems and evaluate 19 state-of-the-art LLMs, finding that UniCode poses a substantial challenge.

**Strengths:**

The motivation for automatically generating high-quality and contamination-resistant algorithmic coding problems and test cases to overcome the limitations of static, human-authored benchmarks is convincing.

The evaluation covers 19 LLMs, which is intensive.

**Weaknesses:**

Test generation metrics: only coverage and correctness are adopted. The fault detection capability of the tests is not investigated, which is quite important: a correct test case with a high coverage can be useless in detecting bugs if the test oracle is weak.

The abstract motivates the work from two aspects: data contamination and limited scalability. I did not find data contamination analysis in this paper, it is therefore difficult to judge the superiority of the proposed benchmark in terms of data contamination mitigation. A helpful solution is to compare the performance of LLMs in newly generated problems and old existing competitive problems with similar difficulty, and compare their results. If LLM's performance is significantly lower on the generated problems, if will demonstrate the consequence of data contamination and further motivate the proposed approach.

The paper has many arbitrary choices without sufficient justification and explanation. For example, why did you select single problem extension, same-type fusion, and cross-type fusion? Why not other strategies? Why did you assemble a final test suite S of 50 cases
with a fixed composition: 20 random, 20 adversarial, and 10 LLM-synthesized inputs, why this distribution?

The approach adopts o4-mini-medium to generate a set of candidate problems, which may cause bias towards o4-mini's performance in comparison to other LLMs.

**Questions:**

How are the three strategies (single problem extension, same-type fusion, and cross-type fusion) selected?

How do the generated problems compare to the existing competitive problems with similar difficulty when running against different LLMs?

---

> ### Author Response · Authors · 2025-11-19
>
> Thank you for your detailed questions. I will address them one by one below.
> > Q1. Test generation metrics: only coverage and correctness are adopted. The fault detection capability of the tests is not investigated, which is quite important: a correct ​test case​ with​ a high coverage can be useless in detecting bugs if the test oracle is weak. ​
>
> Thank you for your question. There appears to be a misunderstanding: **in our paper “coverage” is defined as ​fault-detection coverage​** (the ability of a test suite to reject incorrect solutions)​. See §3.3 (“Evaluating test case quality”) where `Cov@N` is defined as the fraction of incorrect submissions rejected by the test suite.
>
> \begin{equation}
> \text{Cov@N} = \frac{|\{s \in S_{\text{incincorrect}} \mid \text{pass}(s,M)=0 \,\}|}{|S_{\text{incorrect}}|}
> \end{equation}
>
> ---
>
> > Q2. I did not find data contamination analysis in this paper, it is therefore difficult to judge the superiority of the proposed benchmark in terms of data contamination mitigation. A helpful solution is to compare the performance of LLMs  in newly generated problems and old existing competitive problems with similar difficulty, and compare their results. If LLM's performance is significantly lower on the generated problems, it will demonstrate the consequence of data contamination and further motivate the proposed approach.
>
> Thank you for your thoughtful suggestions. Actually, **Section 4.3 already provides the data-contamination analysis.** Specifically, as demonstrated in Line399-401: we constructed three problem sets derived from 50 problems in​**​ LiveCodeBench v1.2**​, an *early version* that carries **potential data-contamination risk** (the leaderboard marks contamination risk in red https://livecodebench.github.io/leaderboard.html). We selected models that are marked as contaminated claude-3.5 and qwen3-235b-a22b, and used our UniCode generation procedure to synthesize new problem variants from these contaminated seeds.
>
> From Figure 3, we observed a **substantial performance drop on UniCode-generated problems versus the original seeds** (​**Claude: 0.70 → 0.21; Qwen3: 0.89 → 0.58**​). As you mentioned, “If LLM's performance is significantly lower on the generated problems, it will demonstrate the consequence of data contamination and further motivates the proposed approach.” Thus, the large gap between seeds and their UniCode variants shows that UniCode produces truly novel problems and offers a stronger test of LLMs' generalization.
>
> We thank the reviewer for this suggestion. We have revised Section 4.3 to provide a clearer interpretation of the contamination analysis.
>
> ---
>
> > Q3. The paper has many arbitrary choices without sufficient justification and explanation. For example, why did you select single problem extension, same-type fusion, and cross-type fusion? Why not other strategies?
>
> These are not arbitrary choices. ​**We have tried different generation strategies**​, as stated in line 102-104 "The key idea behind generating new problems is to modify existing ones. We first observe that naive superficial changes to problems, **such as changing the background story or adding distracting information, ​**do not produce challenging or novel tasks, as confirmed by our experiments (Figure 3)." Specifically:
>
> * Example 1: we converted a card-game queue/stack simulation into an operating-system scheduling scenario (different application narrative, same queue-management logic).
> * Example 2: we wrapped a simple algorithmic problem in a long story or added irrelevant tables/rules that have no effect on the underlying algorithm.
>
> ​**These text-based transformation strategies produce “shadow” questions whose model performance remains essentially unchanged**​ (see Figure 3: shadowQS vs SeedQS). Models still recognize and reproduce memorized solutions for these shadow variants, which means these strategies do not mitigate contamination.
>
> By contrast, UniCode’s three strategies intentionally alter algorithmic structure and complexity:
>
> 1. Single-problem extension — increase algorithmic complexity (e.g., Two Sum → Three Sum).
> 2. Same-type fusion — combine two problems with the same algorithmic tag to create a new variant with similar logic but different interactions.
> 3. Cross-type fusion — merge problems across categories to form genuinely hybrid, more complex challenges.
>
> Empirically, the UniCode variants created by these three strategies show a meaningful performance gap relative to seeds, which indicates novelty and value for evaluation. We are open to further discussion if you have alternative strategies to propose.

---

> ### Author Response · Authors · 2025-11-19
>
> > Q4. The approach adopts o4-mini-medium to generate a set of candidate problems, which may cause ​bias towards o4-mini's performance in comparison to other ​*​​*LLMs*​*.
>
> We are aware of this potential concern and explicitly stated it as a limitation in the conclusion “A limitation of our study is its reliance on a single powerful closed-source LLM, o4-mini, which may introduce bias by generating problems that align with its own strengths”.
>
> To examine this effect, we additionally used the open-source model DeepSeek-R1 to generate 104 problems across five tags and evaluated multiple models with different capability levels. The results show no clear evidence that a model performs substantially better on problems it generated itself. **Compared with human-designed problems and those generated by o4-mini, the ranking of ​**​**LLM**​**​ performance remains consistent. ​**Statistically, the Unicode benchmarks produced by different generators are highly aligned **Pearson r = 0.984, p = 4e-4** (Unicode from deepseek vs. Unicode from o4-mini).
>
> | model                     | Unicode (deepseek-generated) | Unicode (o4mini-generated) | LiveCodeBench (human, no contamination) |
> | --------------------------- | ------------------------------ | ---------------------------- | ----------------------------------------- |
> | gpt-5-2025-08-07          | 0.725                        | 0.677                      | —                                      |
> | o4-mini-2025-04-16        | 0.702                        | 0.661                      | 0.742                                   |
> | deepseek-r1-0528          | 0.616                        | 0.556                      | 0.731                                   |
> | o3-mini-2025-01-31        | 0.51                         | 0.551                      | 0.63                                    |
> | gpt-4.1-mini-2025-04-14   | 0.413                        | 0.424                      | 0.532                                   |
> | google/gemma-3-27bit | 0.146                        | 0.131                      | —                                      |
>
> Different generators may favor slightly different difficulty distributions, which affects absolute scores to some degree, but does not materially change model ranking or the main conclusions. Therefore, to further mitigate generator-source risk, we will release **Unicode-Multi ​**after the rebuttal period. It is a consolidated benchmark co-created by multiple state-of-the-art LLMs (including o1-mini, GPT-4o, Gemini-2.0-Pro, DeepSeek-R1, and Qwen-32B). Each model contributed approximately 100 problems to ensure greater diversity. We have added a detailed analysis of generator bias in Section 4.5.
>
> ---
>
> > Q5. Why did you assemble a final test suite S of 50 cases with a fixed composition: 20 random, 20 adversarial, and 10 LLM-synthesized inputs, why this distribution?
>
> We selected the 20/20/10 composition based on an empirical sweep across multiple distributions, evaluating correctness and coverage over 48 problems and 960 human-crafted solutions. Representative results include:
>
> * (5, 5, 0): correctness 97.9, coverage 77.0
> * (10, 10, 5): correctness 95.8, coverage 81.4
> * (20, 20, 10): correctness 94.0, coverage 87.5 ← selected
> * (30, 30, 20): correctness 91.7, coverage 88.7
>
> From these results, we observed that (a) too few test cases significantly weakens fault detection, (b) too many increases sensitivity to single-case noise and pipeline cost, (c) 20/20/10 balances detection power and robustness while controlling cost. We include these experiments in Appendix D.
>
> We also validated the necessity of each test-case category via ablation. The table shows that every type of test case matters. If we remove any category, the overall performance increases. Overall, the 20/20/10 mix  is a carefully calibrated balance that ensures comprehensive evaluation across both common and edge-case scenarios, maximizing both fault detection and suite reliability.
>
> | Model   | AdvPass | RandPass | LLM-syn Pass | Remove Adv | Remove Rand | Remove LLM-syn | OverallPass |
> | --------- | --------- | ---------- | -------------- | ------------ | ------------- | ---------------- | ------------- |
> | Average | 52.99%  | 55.92%   | 55.53%       | 49.94%     | 44.53%      | 51.21%         | 38.99%      |

---

> > ### Author Response · Authors · 2025-11-19
> >
> > > Q6. How are the three strategies (single problem extension, same-type fusion, and cross-type fusion) selected?
> >
> > First, our goal is to generate diverse coding problems. Inspired by biology, where diversity of life is driven by two key mechanisms: gene mutation and genetic recombination (Baake & Gabriel, 2000; Charlesworth et al., 2009), as noted in line 46. UniCode adopts three complementary strategies to evolve tasks from a seed set.
> >
> > Second, similar approaches have been shown to be effective in other research domains, as discussed in related work (Pei et al., 2025; Huang et al., 2025; Wu et al., 2021).
> >
> > Finally, as detailed in Q3, we conducted extensive evaluations to assess different generation strategies. The results confirm that these three methods are effective in producing diverse problems that are also resilient to data contamination.
> >
> > > Q7. How do the generated problems compare to existing competitive problems of similar difficulty when evaluated on different LLMs?
> >
> > As discussed in Section 4.3, our new problems are generated from the data-contaminated LiveCodeBench-v1 problems. Since they are derived from these seeds, they assess similar knowledge points and maintain comparable difficulty levels. We ran four LLMs of varying levels on both the original seed sets and the UniCode sets. The performance gap is evident: **model performance drops by 35.5% on average when evaluated on UniCode. ​**This drop demonstrates both the presence of data contamination in the original benchmarks and the effectiveness of our benchmark in mitigating it.

---

### Author Response · Authors · 2025-11-28
**Follow-up and Consolidated Response to Reviewers' Common Concerns**

Thank you again for your time and thoughtful review of our paper. We noticed that we haven’t heard back after our rebuttal, and we wanted to check whether there might still be some unresolved concerns. To address this, we would like to first provide a unified response that encompasses common concerns, then summarize our contributions.

> ### Data Contamination Analysis and Need for Generative Evaluation

The reviewers expressed concerns about the effectiveness of UniCode in mitigating data contamination. Our work is fundamentally motivated by this issue, which causes evaluation to shift focus from genuine generalization to mere memorization.

**Our Evidence (Section 4.3):**
We specifically ran a targeted experiment on this. We took 50 problems known to be contaminated in the training data (the 'seed' problems). We then used UniCode to generate novel variations from these seeds.
- The Big Drop: The results were dramatic. Models that scored high on the contaminated seeds showed a performance drop of over 30% on our generated variants (e.g., Claude dropped from 0.70 to 0.21, and Qwen3 from 0.89 to 0.58).
- The Takeaway: This drop is our strongest proof. It shows that even if the seed problems are contaminated, UniCode produces truly novel problems that require models to rely on genuine algorithmic reasoning, not just pattern matching or memory. Our method forces the evaluation back on the right track: generalization.

**The Need for Generative Evaluation**:
As LLMs continue to scale in size, the saturation of benchmarks can be shockingly fast: an Olympic-level difficult problem set AIME can be 98.7% saturated within months of its release. Human experts are slow and expensive, and their data get contaminated quickly. UniCode offers the necessary escape: a generative evaluation framework that can produce an endless stream of structurally novel problems quickly and affordably. This dynamic approach guarantees we are testing generalization, not just memory.


> ### Justification of Problem Generation Strategies

Our strategies were chosen for their effectiveness in generating **structural novelty**. We first experimented with **naive, superficial generation** (termed "Shadow Questions"), like changing the background story or adding distracting information, but found model performance was unchanged relative to the seeds, meaning they did not mitigate contamination. In contrast, UniCode's three core strategies are designed to intentionally alter the algorithmic structure:
- Single-type mutation: changes the core reasoning logic (e.g., Two Sum $\rightarrow$ Three Sum).
- Same-Type Fusion: combines problems with similar logic but different interactions.
- Cross-Type Fusion: merges problems across categories to form genuinely hybrid, complex challenges.
The resulting observed performance drop (over 30%) on UniCode variants directly justifies that these methods yield structurally novel problems beyond mere paraphrases. **This also explains why we didn't pursue other, more comprehensive generation strategies: if LLMs already generalize well in some areas, testing them there lacks discriminative power and can't guide meaningful development.**


> ### Addressing Generator Bias (o4-mini)

The concern about using a single generator (o4-mini) potentially biasing the benchmark is very valid, which is why we listed it as a limitation and followed up with more experiments.
- **New Experimental Results (Section 4.5)**:
To check this, we re-ran tests using problems generated by a completely different model, **DeepSeek-R1**. We found no clear evidence that any model (including o4-mini or DeepSeek-R1) performed disproportionately better on the problems it generated itself. When comparing performance on DeepSeek-generated vs. o4-mini-generated sets, **the model ranking remained identical**. The statistical correlation between the two sets is extremely high: Pearson $r=0.984$ ($p<0.001$). While different generators may create problems with slightly different difficulty levels (affecting absolute scores), they do not change the relative capabilities of the tested LLMs.
- **Transparency and Future Mitigation**:
Furthermore, while human-authored datasets have their own opaque biases, our method is fully reproducible, allowing for the tracing and estimation of any potential bias, which offers a significant advantage in transparency. To further mitigate this risk, we are actively preparing **UniCode-Multi**, a consolidated benchmark co-created by multiple state-of-the-art LLMs (including O1-mini, GPT-4o, Gemini-2.0-Pro, DeepSeek-R1, and Qwen-32B). This maximizes diversity and alleviates single-source bias.

---

> ### Author Response · Authors · 2025-11-28
>
> > ### Key Contributions of UniCode
>
> Our work highlights the potential data-contamination risk of current evaluation methods and proposes a scalable, reliable, and contamination-resistant evaluation framework.
> - We introduce biologically-inspired evolutionary strategies—single-extension, same-type fusion, and cross-type fusion—to generate structurally novel problems. This prevents memorization and causes an average performance drop of over 30% on contaminated seed problems.
> - We propose a stress-driven test case algorithm that achieves 94.5% accuracy without a canonical solution, serving as the cornerstone for reliably generating competitive coding problems at scale.
> - UniCode's average cost is $0.041 and 8.2 minutes per problem, drastically undercutting the ~40 hours and hundreds of dollars required for human-authored tasks, which are quickly contaminated.This superior scalability and cost-effectiveness serves as the cornerstone for truly testing generalization performance.
>
>
> ### **We are eager to receive your updated assessment and remain available to clarify any new or outstanding questions you may have.**

---

### Meta-Review · Area_Chair_XkYG · 2025-12-06

**Summary:**

The reviewers' concerns can be summarized as follows:

- This benchmark cannot solve the issue of vast memorization of training data
- The paper does not discuss or compare with Evol-Instruct
- The entire framework relies heavily on o4-mini for both task generation and final adjudication
- The framework depends on seed problems from TACO
- Limited ablations
- Quality assurance remains narrow

**Reviewer Concerns:**

Well addressed:
- Test generation metrics: only coverage and correctness are adopted
- The paper does not discuss or compare with Evol-Instruct
- The entire framework relies heavily on o4-mini for both task generation and final adjudication

Still outstanding:
- The framework depends on seed problems from TACO
- Limited ablations
- Quality assurance remains narrow

For these concerns I think still outstanding, the common key issue is:
- The definition of "vast memorization of training data" is fuzzy. This paper uses LLM's accuracies as a metric to judge whether a dataset suffers from the issue of vast memorization of training, but this definition does not convince the reviewers. This paper can prove that the generated problems are harder, but lack of clear analysis where the hardness is from.

**Reviewer Scores:**

According to the reviewer concern summary above, I guess:
- Reviewer tfss will raise the score from 4 to 6
- Reviewer bwy7 will keep the score (4)
- Reviewer 2L99 will keep the score (4)
- Reviewer xybC will keep the score (6)

---

### Decision · Program_Chairs · 2026-01-26

Reject